# Therapeutic Frontiers in Gastroesophageal Cancer: Contemporary Concepts in Management and Therapy

**DOI:** 10.3390/ijms262311424

**Published:** 2025-11-26

**Authors:** Supriya Peshin, Ehab Takrori, Naga Anvesh Kodali, Faizan Bashir, Michael Gibson, Sakshi Singal

**Affiliations:** 1Norton Community Hospital, Norton, VA 24273, USA; 2College of Medicine, Alfaisal University, Riyadh 11533, Saudi Arabia; 3HCA Florida Ocala Hospital, University of Central Florida, Ocala, FL 34471, USA; 4School of Medicine, Shiraz University of Medical Sciences, Shiraz 71348-45794, Iran; 5Vanderbilt-Ingram Cancer Center, Nashville, TN 37232, USA; 6Department of Hematology & Oncology, East Tennessee State University, Johnson City, TN 37614, USA

**Keywords:** gastroesophageal cancer, immunotherapy, *HER2*-positive adenocarcinoma, perioperative chemotherapy, targeted therapy, biomarkers

## Abstract

Gastroesophageal cancer (GEC) represents a global health burden, with rising incidence and high mortality. Despite advancements in early detection and systemic therapies, outcomes remain poor, especially in advanced stages. Management requires a multidisciplinary, multimodal approach that integrates surgery, chemotherapy, radiotherapy, targeted agents, and immunotherapy, tailored by tumor histology, location, and molecular profile. For localized disease, perioperative chemotherapy or chemoradiotherapy is standard, with adjuvant immunotherapy now emerging in selected high-risk cases. In metastatic or unresectable settings, systemic therapy forms the backbone of treatment, with biomarker-driven regimens targeting *HER2*, PD-L1, *MSI-H/dMMR*, and *CLDN18.2*, offering improved outcomes. Novel agents and combinations, including bispecific antibodies, *FGFR2* inhibitors, and immunotherapy-based strategies, are actively being explored in clinical trials. This review provides a comprehensive overview of the evolving therapeutic landscape of GEC. It emphasizes the growing role of precision medicine and the integration of emerging clinical data into practice.

## 1. Introduction

Gastroesophageal cancers (GECs), including gastric, esophageal, and gastroesophageal junction (GEJ) adenocarcinomas (ACs) and squamous cell carcinomas (SCCs), remain among the leading causes of cancer-related mortality worldwide [1]. Globally, gastric and esophageal cancers accounted for approximately 1.6 million new cases and 1.3 million deaths in 2020, with esophageal cancer alone contributing an estimated 11.7 million disability-adjusted life years (DALYs) in 2019, underscoring the substantial global health burden posed by these malignancies [2,3]. These cancers are frequently diagnosed at advanced stages and exhibit aggressive behavior, resulting in poor long-term outcomes [4].

While therapeutic advances have emerged in recent years, including biomarker-informed approaches and the integration of targeted agents and immunotherapies, the overall prognosis remains guarded. Heterogenous tumor biology and geographic variation in disease patterns further contribute to diverse clinical outcomes [4,5].

The management of GEC increasingly relies on stage-specific multimodal strategies informed by histology and molecular biomarkers, with recent advances in systemic therapy, immunotherapy, and targeted agents reshaping treatment across both localized and metastatic settings [6,7,8,9,10,11].

In contrast to prior reviews that typically address isolated therapeutic modalities or specific disease stages, this work integrates the full spectrum of localized, advanced, and emerging treatment strategies, incorporating the most recent phase II/III trial data and biomarker-driven approaches. By combining contemporary clinical evidence with evolving translational tools such as ctDNA monitoring, AI-driven analytics, and nanoscale imaging technologies, this review provides a comprehensive and forward-looking synthesis that extends beyond the scope of existing literature.

## 2. Management of Localized GEC

The management of localized GEC has evolved significantly in recent years, with multidisciplinary strategies now forming the backbone of treatment. The overarching goal is curative intent through a combination of surgery, systemic therapy, and, in selected cases, radiotherapy, guided by tumor location, histology, and clinical stage.

### 2.1. Initial Workup and Staging

Accurate staging is essential to determine the treatment strategy and prevent missed metastatic disease. The standard workup includes upper endoscopy with biopsy, which confirms malignancy, identifies the histologic subtype (AC versus SCC), and maps the tumor location and extent within the esophagus, gastroesophageal junction, or stomach [12]. Endoscopic ultrasound (EUS) provides optimal assessments of tumor depth invasion (T stage) and regional lymph node involvement (N stage). These parameters are crucial for determining whether patients require upfront surgery, neoadjuvant therapy, or a combined approach (Figure 1) [13,14].

CT of chest, abdomen, and pelvis with PET-CT evaluates for distant metastatic diseases involving the liver, lungs, or non-regional lymph nodes. PET imaging offers enhanced sensitivity for detecting occult metastases that conventional CT may not visualize [15,16]. Diagnostic laparoscopy is frequently employed in gastric and GEJ malignancies, particularly with advanced locoregional diseases. This procedure detects microscopic peritoneal diseases that may be radiographically occult [17,18].

These comprehensive staging modalities collectively guide therapeutic decision-making, determining whether patients should proceed directly to surgical resection, receive neoadjuvant therapy, or be managed with palliative intent when metastatic disease is identified. Multidisciplinary tumor board evaluation remains essential for tailoring individualized therapy based on this staging information.

### 2.2. Surgical Resection

Surgical resection remains the cornerstone of curative intent treatment for localized GECs. The choice of surgical approach is determined primarily by tumor location, histology, and extent of disease. Equally important is the surgical setting where outcomes are significantly improved when procedures are performed in high-volume centers with dedicated expertise in upper gastrointestinal malignancies. For distal gastric cancers, a subtotal (distal) gastrectomy with adequate proximal and distal margins is typically sufficient, provided the proximal extent of the tumor does not involve the cardia. For GEJ tumors, particularly Siewert types I and II, either transhiatal or transthoracic esophagectomy may be performed, with the choice guided by tumor length, local invasion, and institutional expertise. For esophageal tumors, esophagectomy with either via an Ivor Lewis (two-field), McKeown (three-field), or transhiatal approach is standard being tailored to tumor location and patient factors [19,20,21,22,23].

Lymphadenectomy is a critical component of curative surgery. A D2 lymph node dissection, which includes removal of perigastric and adjacent regional lymph nodes (stations 1–11), is standard in many countries, especially in East Asia where it is associated with improved staging accuracy and survival. In Western centers, modified D2 dissections are often performed due to concerns regarding morbidity, though outcomes have improved with surgical standardization and centralization of care [24,25,26].

A recent network meta-analysis by Cai et al. confirmed the superiority of perioperative chemotherapy regimens over surgery alone, with FLOT (fluorouracil, leucovorin, oxaliplatin, and docetaxel) emerging as the most favorable option based on overall survival (OS) benefit among eight evaluated protocols [27].

### 2.3. Multimodality Therapy

#### 2.3.1. Perioperative Chemotherapy

Perioperative chemotherapy represents the standard of care for patients with resectable gastric and GEJ ACs, particularly those with clinical stage T2 or higher or with radiologic or pathologic evidence of nodal involvement. This approach aims to enhance resectability by downstaging the tumor, eradicate micrometastatic disease early, and improving both disease-free and OS.

The FLOT regimen, comprising 5-fluorouracil (2600 mg/m^2^ as a 24-h infusion step), leucovorin (200 mg/m^2^), oxaliplatin (85 mg/m^2^), and docetaxel (50 mg/m^2^) administered every two weeks, is the current standard in fit patients. This regimen was established by the phase III FLOT4-AIO trial, which demonstrated a significant survival benefit compared to the older ECF/ECX regimens. In that trial, median OS improved from 35 months with ECF/ECX to 50 months with FLOT, with a higher rate of R0 resection and better pathological response [28,29].

The treatment plan typically consists of four cycles of neoadjuvant FLOT, followed by surgical resection with curative intent and then four additional postoperative cycles, assuming adequate recovery and performance status. Adherence to both pre- and post-operative components is ideal, although postoperative therapy completion rates can be limited by surgical recovery and toxicity [29].

Given the intensity of perioperative chemotherapy regimens, patient selection is critical. Candidates should have a good performance status (Eastern Cooperative Oncology Group [ECOG] 0–1), preserved organ function, and adequate nutritional status to tolerate treatment. In frail patients or those with significant comorbidities, dose-modified regimens or alternative combinations such as CAPOX (capecitabine and oxaliplatin) may be considered. However, these are generally reserved for palliative settings or for patients in whom curative-intent therapy may not be feasible due to medical limitations.

This perioperative strategy is endorsed by major international clinical guidelines, including those from the National Comprehensive Cancer Network (NCCN), the European Society for Medical Oncology (ESMO), and the American Society of Clinical Oncology (ASCO), as the standard of care for patients with resectable gastric and GEJ AC. By integrating systemic therapy with surgery, this approach aims to improve R0 resection rates, downstage tumors, and address micrometastatic disease early. Consequently, it has largely replaced the historical surgery-first paradigm in Western oncology practice, where up-front resection alone is now considered suboptimal for most patients.

The phase III MATTERHORN trial, led by Janjigian et al., evaluated the addition of perioperative durvalumab, an anti–PD-L1 antibody, to standard chemotherapy in patients with resectable gastric and gastroesophageal junction adenocarcinoma. This multinational, double-blind study demonstrated a significant improvement in event-free survival for patients receiving durvalumab, with no adverse impact on surgical outcomes. These results support the incorporation of immunotherapy into perioperative regimens and may shift the treatment paradigm for localized disease [30].

Interpretation of perioperative immunotherapy trials requires caution due to significant heterogeneity in study designs and patient populations. FLOT4 established a strong chemotherapy backbone; however, its widespread use outside Western populations remains limited. MATTERHORN enrolled predominantly Western patients, raising questions about generalizability to East Asian populations where tumor biology, surgical quality, and perioperative management differ. Furthermore, PD-L1 assessment methodologies varied across these trials, complicating cross-study comparisons. These factors must be considered when integrating perioperative immunotherapy into routine practice.

#### 2.3.2. Neoadjuvant Chemoradiotherapy

Neoadjuvant chemoradiotherapy (nCRT) is the preferred approach for patients with resectable distal esophageal and Siewert type I GEJ tumors, particularly in the setting of squamous cell carcinoma (SCC) or high-risk AC. This strategy is supported by robust evidence demonstrating improved survival and local control when compared to surgery alone.

The CROSS trial (Chemoradiotherapy for Esophageal Cancer Followed by Surgery Study) established the current standard regimen, which consists of weekly carboplatin and paclitaxel (50 mg/m^2^) for five weeks, administered concurrently with radiation therapy to a total dose of 41.4 Gy in 23 fractions. This regimen was followed by surgical resection 4–6 weeks after completion of chemoradiation [31].

In the CROSS trial, this approach significantly improved R0 resection rates, median OS, and pathologic complete response (pCR), particularly notable in patients with squamous cell histology, where pCR rates exceeded 45%. ACs also benefited, with improved survival and reduced locoregional recurrence, although to a lesser extent than SCC.

The recently published ESOPEC trial provided the first direct phase III comparison of perioperative chemotherapy and neoadjuvant chemoradiotherapy in resectable esophageal and gastroesophageal junction adenocarcinoma. As summarized by Modi et al., perioperative FLOT achieved significantly superior overall and disease-free survival rates compared with the CROSS regimen, supporting a paradigm shift toward systemic perioperative therapy as the preferred approach in this population. These findings strengthen the rationale for integrating immune checkpoint inhibitors with FLOT-based regimens in ongoing perioperative studies such as DANTE and METTERHORN [32].

Although ESOPEC suggests superiority of perioperative FLOT over nCRT with CROSS, direct comparisons are limited by differences in staging modalities, radiotherapy techniques, and surgical expertise across participating centers. CROSS was conducted in high-volume institutions with standardized radiation quality assurance, whereas perioperative chemotherapy trials exist within varied real-world surgical quality environments. These discrepancies introduce potential confounders that warrant careful interpretation before universal adoption of either approach.

This trimodality approach is generally reserved for tumors located in the mid-to-distal esophagus and GEJ, where radiation planning can safely encompass the primary tumor and regional lymphatics. It is now considered standard of care by major guidelines, including NCCN and ESMO, for appropriate histologies and anatomical locations.

Kridis et al. pooled data from two phase III RCTs (*n* = 727) and found that perioperative chemotherapy improved OS (HR = 0.72, 95% CI: 0.55–0.95) and disease-free survival (DFS) (HR = 0.65, 95% CI: 0.50–0.85) compared to surgery alone. In the CLASSIC trial, adjuvant chemotherapy also showed benefit (OS: HR = 0.72, CI: 0.52–1.00; DFS: HR = 0.56, CI: 0.44–0.72). Adjuvant chemoradiotherapy was superior to surgery alone (HR = 1.35, 95% CI: 1.09–1.66; *p* = 0.005) [33].

### 2.4. Adjuvant Therapy

The role of adjuvant therapy in localized GEC is shaped by the initial treatment approach and the patient’s postoperative recovery and pathologic findings. For patients treated with perioperative chemotherapy, such as the FLOT regimen, completion of the four planned postoperative cycles is recommended to maximize survival benefit [29]. However, postoperative tolerance is often limited due to surgical morbidity, nutritional deficits, or functional decline, and fewer than half of patients in real-world settings complete the full course. Nonetheless, efforts should be made to resume therapy in eligible patients, as data from the FLOT4 trial suggest improved outcomes with treatment completion [34].

In patients who undergo neoadjuvant chemoradiotherapy followed by surgery, most commonly for esophageal or Siewert type I GEJ cancers, the presence of residual pathologic disease (i.e., no complete response) identifies a high-risk population with significant recurrence risk. In this setting, adjuvant immunotherapy with nivolumab for up to one year has become new standard of care, based on the results of the CheckMate 577 trial as shown in Figure 2. This phase III trial demonstrated a significant improvement in DFS with adjuvant nivolumab compared to placebo (22.4 vs. 11.0 months), with benefit observed across histologic subtypes and PD-L1 expression levels [35,36].

### 2.5. Emerging Therapies and Trials

The treatment landscape for localized GEC is rapidly evolving, with multiple ongoing trials exploring the integration of immunotherapy and biomarker-driven strategies into curative-intent regimens (Table 1).

Checkpoint inhibitor-based combinations are at the forefront of investigation. Trials such as DANTE (NCT03421288), VESTIGE (NCT03443856), and NeoFLOT (NCT02872103) are evaluating the safety and efficacy of adding immune checkpoint inhibitors (ICIs), such as nivolumab or atezolizumab, to perioperative chemotherapy (e.g., FLOT) in patients with resectable gastric and GEJ ACs. Preliminary data suggest potential for improved pathologic response and long-term survival, though definitive results are awaited. These studies represent a shift toward leveraging the immunogenic potential of chemotherapy to prime the tumor microenvironment for immune activation [37,38,39].

In parallel, biomarker-driven therapy is gaining relevance even in the localized setting. *HER2* testing is already standard in metastatic disease but is now being explored for its predictive value in early-stage tumors and potential role in perioperative *HER2*-targeted strategies [40]. Similarly, PD-L1 expression and microsatellite instability (MSI) or mismatch repair deficiency (dMMR), which have established roles in metastatic immunotherapy are being investigated for integration into neoadjuvant or adjuvant immunotherapy trials. The identification of MSI-high status may eventually guide the use of checkpoint blockade in the curative setting, independent of chemotherapy [41,42].

These emerging paradigms reflect a broader shift toward personalized treatment strategies, aiming to refine therapy based on tumor biology, treatment response, and molecular profiling. Future trial outcomes are likely to redefine the standard of care and further individualize the management of localized GEC.

### 2.6. Follow-Up

Post-treatment surveillance for patients with localized GEC focuses on early detection of recurrence, management of complications, and support for nutritional and functional recovery. While no standardized follow-up protocol exists, practices are typically shaped by expert guidelines and institutional experience. During the first two years when recurrence risk is highest, clinical evaluations including physical exams and symptom reviews, are generally scheduled every 3 to 6 months. After this period, visits may be spaced to annual intervals depending on individual risk. Imaging, especially contrast-enhanced CT scans of the chest, abdomen, and pelvis, is used based on clinical need, particularly in high-risk or symptomatic patients; however, its routine use in asymptomatic individuals remains debated. Endoscopic surveillance is considered for select patients, especially those with esophageal or GEJ cancers. Laboratory testing, including nutritional and tumor markers, may be used selectively, though they are not standard in all follow-up protocols [43].

### 2.7. Key Takeaways

FLOT remains the preferred perioperative regimen for fit patients with localized gastric and GEJ AC, based on FLOT4 data demonstrating superior survival and pathologic response compared to ECF/ECX.The CROSS protocol (carboplatin/paclitaxel + 41.4 Gy) is standard for distal esophageal and Siewert type I GEJ tumors, particularly with squamous histology.Adjuvant nivolumab has become the new standard for patients with residual disease after neoadjuvant chemoradiotherapy, per CheckMate 577.

What We Still Don’t Know

Is FLOT equally effective across all histologies?A retrospective multicenter study suggests that patients with signet-ring cell positive gastric cancer have shorter DFS and OS when treated with perioperative FLOT compared to surgery followed by adjuvant chemotherapy [44]. This raises concerns about the suitability of FLOT for this subgroup and warrants further prospective evaluation.What is the clinical utility of circulating tumor DNA (ctDNA) in esophagogastric cancers?

A 2024 meta-analysis by Zhang et al. (1144 patients), found ctDNA positivity associated with poor prognosis in esophageal cancer. Similarly, Gao et al. (1193 GC patients) reported ctDNA positivity correlated with worse survival in gastric cancer. A systematic review by Mi et al. (432 patients) showed that ctDNA positivity both pre- and postoperatively significantly predicted recurrence and reduced OS (HR = 6.37; HR = 4.58). These findings support ctDNA’s potential as a biomarker, though its role in clinical decision-making remains to be validated [45,46,47].

## 3. Management of Metastatic/Unresectable GEC

When GEC spreads beyond the primary site and becomes metastatic, it is no longer considered curable. At this stage, treatment is focused on palliative goals like extending survival, relieving symptoms, and enhancing the patients’ quality of life. Management typically involves systemic therapies such as chemotherapy, immunotherapy, and targeted therapy. Chemotherapy works by attacking rapidly dividing cancer cells throughout the body, while immunotherapy enhances the patient’s immune response against the tumor. Targeted therapies are designed to interfere with specific molecular alterations driving the cancer’s growth. Treatment decisions are increasingly guided by tumor biomarkers, including *HER2* (a growth-promoting protein), PD-L1 (a marker associated with response to ICIs), and *MSI-H/dMMR* status (indicating genetic instability and potential sensitivity to immunotherapy) (Figure 3).

### 3.1. First-Line Treatment

#### 3.1.1. Chemotherapy Alone

Combination chemotherapy with a fluoropyrimidine (e.g., 5-fluorouracil or capecitabine) and a platinum agent (cisplatin or oxaliplatin) has long been the cornerstone of first-line treatment for metastatic GECs, regardless of tumor location or histology [48,49]. Despite its widespread use, median OS with this regimen remains under one year. Systemic therapy aims to relieve symptoms, improve quality of life, and extend survival. Evidence consistently supports a survival benefit with chemotherapy, and platinum-based combinations remain the first-line standard for advanced disease [50,51].

#### 3.1.2. Chemoimmunotherapy (*HER2-*Negative Disease)

For *HER2*-negative advanced gastroesophageal cancers, first-line treatment is largely guided by PD-L1 expression. Multiple phase III trials have established chemoimmunotherapy as a standard option for unresectable or metastatic disease, demonstrating consistent survival benefits with PD-1 inhibitors added to chemotherapy [52,53,54,55,56,57,58].

Across pivotal global trials, including ATTRACTION-4, KEYNOTE-859, RATIONALE-305, and CheckMate 649, PD-1 blockade combined with chemotherapy improved outcomes in HER2-negative gastric and GEJ adenocarcinoma, with the greatest benefit observed in tumors with higher PD-L1 expression. These trials uniformly support the role of chemoimmunotherapy as first-line treatment, while detailed efficacy results are summarized in Table 2. CheckMate 648 and KEYNOTE-590 further expanded this benefit to esophageal squamous cell carcinoma, establishing PD-1 inhibitors as integral to first-line therapy in biomarker-selected ESCC [56,57,58,59].

Based on this evidence, ASCO guidelines recommend nivolumab plus chemotherapy for gastric, GEJ, or esophageal adenocarcinoma with PD-L1 CPS ≥ 5; pembrolizumab plus chemotherapy for CPS ≥ 10 in adenocarcinoma or ESCC; and either nivolumab plus chemotherapy or nivolumab plus ipilimumab for ESCC with TPS ≥ 1 [60].

It is noteworthy that major immunotherapy trials differ markedly in geographic representation and PD-L1 assay methodology. For example, ATTRACTION-4, conducted exclusively in Asia, did not show an OS advantage, likely due to high post-progression immunotherapy use, whereas CheckMate 649 and KEYNOTE-859 enrolled more diverse global populations but applied different PD-L1 scoring thresholds. Such geographic and methodological heterogeneity complicates direct cross-trial comparisons and underscores the need for harmonized predictive biomarker strategies.

#### 3.1.3. *HER2-*Targeted Therapy (*HER2-*Positive Disease)

The ToGA trial, a pivotal phase III study, established *HER2* as a predictive biomarker in gastric and GEJ adenocarcinoma. Adding trastuzumab to chemotherapy improved overall survival compared with chemotherapy alone (13.8 vs. 11.1 months; HR 0.74; *p* = 0.0046), with a later updated analysis reporting OS of 13.1 vs. 11.7 months (0.80) [61].

The KEYNOTE-811 trial evaluated pembrolizumab added to trastuzumab plus fluoropyrimidine- and platinum-based chemotherapy in previously untreated HER2-positive gastric and GEJ adenocarcinoma. The addition of pembrolizumab significantly increased the objective response rate (74.4% vs. 51.9%; *p* = 0.00006) and complete response rate (11.3% vs. 3.1%) compared with trastuzumab-chemotherapy alone, establishing this triplet as the preferred first-line regimen in eligible patients [60,62].

Interpretation of HER2-directed trials is complicated by variability in HER2 testing methodologies across regions. Differences in scoring criteria for gastric versus breast cancer, interlaboratory heterogeneity, and limited confirmatory testing contribute to inconsistent HER2 classification, directly affecting trial eligibility and potentially explaining variations in response rates across global studies.

#### 3.1.4. MSI-H/dMMR-Directed Immunotherapy

Microsatellite instability-high (MSI-H) or mismatch repair-deficient (dMMR) GEJ ACs are characterized by high mutational burden and abundant neoantigen load, features that confer strong sensitivity to immune checkpoint inhibition [63]. Pembrolizumab has received tumor-agnostic approval for unresectable or metastatic MSI-H/dMMR cancers based on sustained responses across multiple pivotal studies.

In KEYNOTE-062, pembrolizumab monotherapy demonstrated non-inferior overall survival compared with chemotherapy in PD-L1 CPS ≥ 1 tumors (OS 10.6 vs. 11.1 months; HR 0.90) and superior survival in CPS ≥ 10 tumors (17.4 vs. 10.8 months; HR 0.62) [64,65]. However, adding pembrolizumab to chemotherapy did not improve OS in these subgroups.

In CheckMate-649, nivolumab plus chemotherapy significantly prolonged OS in HER2-negative gastric, GEJ, and esophageal adenocarcinoma with PD-L1 CPS ≥ 5 (14.4 vs. 11.1 months; HR 0.71), with benefit also seen in the overall population (13.8 vs. 11.6 months; HR 0.80) [56,66,67]. CheckMate-648 expanded these findings to ESCC, with nivolumab plus chemotherapy achieving a median OS of 15.4 months and nivolumab plus ipilimumab achieving 13.7 months, both superior to chemotherapy alone (9.1 months) in PD-L1-expressing tumors [56,57]. Updated follow-up confirmed sustained benefit [58,68].

KEYNOTE-590 demonstrated a marked survival advantage for pembrolizumab plus chemotherapy in advanced esophageal and GEJ adenocarcinoma or squamous cell carcinoma with PD-L1 CPS ≥ 10 (13.5 vs. 9.4 months; HR 0.62) [59].

In the second-line setting, KEYNOTE-061 showed that pembrolizumab did not meet the primary OS endpoint versus paclitaxel in PD-L1 CPS ≥ 1 tumors, though patients with CPS ≥ 10 derived greater benefit [64]. KEYNOTE-059 reported an ORR of 11.6% in heavily pretreated disease, with higher responses in PD-L1-positive tumors (15.5% vs. 6.4%) and durable responses (median DoR 16.3 vs. 6.9 months) [69,70,71,72].

ATTRACTION-2 demonstrated that nivolumab significantly prolonged OS compared with placebo in previously treated advanced gastric cancer (OS 5.3 vs. 4.1 months; HR 0.63) [73], although its applicability remains limited by the exclusively Asian study population. In contrast, JAVELIN Gastric 300 showed no survival advantage for avelumab over chemotherapy (OS 4.6 vs. 5.0 months; HR 1.1) [74].

Despite clear efficacy in MSI-H/dMMR disease, most MSI data in gastroesophageal cancer derive from subgroup analyses rather than dedicated trials. MSI prevalence varies by geography and tumor site, and inconsistencies in MSI testing (PCR, IHC, NGS) further complicate cross-study comparisons, underscoring the need for standardized assessment.

#### 3.1.5. Triple-Negative Cancers (*HER2-*Negative, PD-L1-Low, MSI-Stable)

Patients with triple-negative gastroesophageal cancers (defined as *HER2*-negative, PD-L1-negative, and microsatellite stable/proficient mismatch repair [MSS/pMMR]) do not benefit from immunotherapy or *HER2*-targeted treatments [4]. Chemotherapy, typically a fluoropyrimidine plus a platinum agent remains the standard. Trials are ongoing to identify biomarkers and targeted options for this difficult-to-treat group, including those who are ineligible for trastuzumab deruxtecan or immunotherapy.

Another promising target is fibroblast growth factor receptor 2b (*FGFR2b*). Bemarituzumab, a monoclonal antibody against *FGFR2b*, demonstrated clinical benefit in the FIGHT trial when combined with chemotherapy, improving both PFS and OS compared to chemotherapy alone [75]. Although standard chemotherapy remains the first-line therapy for triple-negative GEJ cancers, ongoing clinical development of *CLDN18.2*- and *FGFR2b*-targeted therapies points toward a future era of biomarker-driven treatment strategies.

### 3.2. Second Line and Beyond

#### 3.2.1. *HER2-*Positive Cancers

For patients with advanced *HER2*-positive gastric or GEJ AC who progress following a trastuzumab-based regimen, trastuzumab deruxtecan is the preferred second-line therapy. This recommendation is based on DESTINY-Gastric01 trial, which demonstrates substantially higher response rates and improved survival with trastuzumab deruxtecan compared with physician’s choice chemotherapy in patients previously treated with at least two lines of therapy [76]. These results supported regulatory approval of trastuzumab deruxtecan for use in the second-line or later setting.

Ongoing evaluations continues in DESTINY-Gastric04, a phase III study comparing trastuzumab deruxtecan with ramucirumab plus paclitaxel, the standard chemotherapy regimen for second-line treatment [77].

#### 3.2.2. *HER2-*Negative Cancers

##### Immunotherapy Options

In *HER2*-negative gastroesophageal cancers, immunotherapy is key in second-line treatment. The KEYNOTE-181 trial compared pembrolizumab to chemotherapy in previously treated esophageal cancer and demonstrated a significant overall survival benefit in patients with PD-L1 CPS ≥ 10 (9.3 vs. 6.7 months; HR 0.69; *p* = 0.0074) [78].

Similarly, ATTRACTION-3 is a phase 3 trial in advanced ESCC aimed at comparing nivolumab to chemotherapy and showed a significant OS benefit regardless of PD-L1 expression [79]. Thus, pembrolizumab (for CPS ≥ 10) and nivolumab (regardless of PD-L1 status) are viable second-line options in ESCC.

##### Chemotherapy Options

For patients ineligible for trastuzumab deruxtecan or immunotherapy, chemotherapy remains the cornerstone of second-line management. The RAINBOW trial established ramucirumab plus paclitaxel as the preferred regimen after progression on first-line therapy, demonstrating improved survival compared with chemotherapy alone [80].

The RAMIRIS trial compared FOLFIRI plus ramucirumab against paclitaxel plus ramucirumab, specifically in patients with prior exposure to docetaxel, such as those treated with FLOT. The FOLFIRI-based regimen yielded a higher objective response rate (25% vs. 8%) and longer PFS [81].

Other chemotherapy options used in second-line or later settings include taxanes (docetaxel, paclitaxel), irinotecan, trifluridine plus tipiracil (TAS-102), and regorafenib [82,83,84,85,86,87].

### 3.3. Third Line and Beyond

Trifluridine and tipiracil (TAS-102) improved median OS compared to placebo in the phase 3 TAGS trial (5.7 vs. 3.6 months; *p* = 0.003), leading to FDA approval for use in this setting [86].

ICIs such as pembrolizumab (assessed in the KEYNOTE-059 trial) [72] and nivolumab (assessed in the ATTRACTION-2 trial) [73] have also demonstrated activity in patients who had received two or more prior lines of therapy.

Regorafenib, a multi-kinase inhibitor, showed promise in the INTEGRATE IIa phase 3 trial (ASCO GI 2023), significantly improving OS in later-line advanced gastroesophageal cancer [88].

Apatinib, a *VEGFR* inhibitor approved by the CFDA, improved PFS and OS in Chinese patients with advanced gastric or GEJ AC [89]. However, no significant OS benefit was observed in Western populations, as shown in the phase III ANGEL trial [90].

Fruquintinib, a selective *VEGFR* inhibitor, was tested with paclitaxel in a phase Ib/II study for second-line metastatic GC [91]. At the recommended phase 2 dose (RP2D), the median PFS was 4 months, and the median OS was 8.5 months, with an ORR of 25.9% and a disease control rate (DCR) of 66.7%

The phase 3 FRUTIGA trial compared fruquintinib plus paclitaxel to placebo plus paclitaxel in 703 Chinese patients with advanced gastric or GEJ AC [92]. PFS significantly improved, though OS did not. ORR and DCR were higher with fruquintinib, with manageable hematologic toxicity and no new safety concerns. Fruquintinib plus paclitaxel emerges as a potential second-line treatment option for Chinese patients with advanced gastric or GEJ AC.

A summary of major completed and ongoing clinical trials in metastatic or unresectable gastroesophageal cancer is presented in Table 2.

### 3.4. Recent Clinical Trials in Metastatic GECs

#### 3.4.1. Zolbetuximab Trials: SPOTLIGHT and GLOW

Zolbetuximab, a monoclonal antibody directed against claudin 18.2 (*CLDN18.2*), has been Zolbetuximab, targeting *CLDN18.2*, was evaluated in two key phase 3 trials. In SPOTLIGHT, zolbetuximab plus mFOLFOX6 significantly improved PFS and OS compared to placebo [93,94].

Similarly, the GLOW trial evaluated zolbetuximab plus CAPOX against placebo plus CAPOX in the first-line treatment of *CLDN18.2*-positive, *HER2*-negative advanced gastric or GEJ AC. Patients receiving zolbetuximab achieved longer PFS and OS compared to the placebo arm [95,96].

Although both SPOTLIGHT and GLOW showed significant benefit with zolbetuximab, *CLDN18.2* expression assays lack global standardization. Thresholds for positivity, variability in antibody clones, and heterogenous staining patterns may impact patient selection in real-world settings. Additionally, both trials enrolled predominantly East Asian participants, raising questions about generalizability to Western populations with different disease biology and treatment pathways.

#### 3.4.2. Zanidatamab: *HER2* Bispecific Antibody

Zanidatamab, a bispecific antibody targeting two distinct *HER2* epitopes, has shown promise as a novel therapeutic option for *HER2*-positive metastatic GEJ AC. In early-phase results from the HERIZON-GEA-01 study, zanidatamab combined with chemotherapy achieved an ORR of 75% and a median PFS of 12.0 months [97].

Updated phase 2 data presented at ESMO 2024 demonstrated further improvement, with zanidatamab plus chemotherapy achieving a median PFS of 15.2 months and a 24-month OS rate of 65%. The confirmed objective response rate (cORR) was reported at 84%, and the regimen maintained a manageable safety profile [98]. A phase 3 study is ongoing to validate these encouraging results against existing *HER2*-targeted standards.

#### 3.4.3. KEYNOTE-859: Pembrolizumab Plus Chemotherapy

The KEYNOTE-859 trial investigated the addition of pembrolizumab to chemotherapy in patients with *HER2*-negative advanced gastric or GEJ AC. Early analyses demonstrated a PFS benefit with the pembrolizumab-containing regimen, although final OS results are still pending [53,99]. The integration of pembrolizumab into standard cytotoxic chemotherapy backbones is anticipated to provide an incremental survival advantage.

#### 3.4.4. LEAP-014: Pembrolizumab and Lenvatinib Combination

The LEAP-014 trial evaluated the combination of pembrolizumab (Keytruda) and lenvatinib (Lenvima) with chemotherapy in patients with metastatic ESCC and gastric cancers. This study aimed to assess the synergistic potential of integrating immune checkpoint inhibition with anti-angiogenic therapy [100]. Although the trial demonstrated improvement in PFS, it did not meet its primary endpoint of extending OS [101], highlighting the ongoing challenges in treating these malignancies and the need for continued therapeutic innovation.

#### 3.4.5. NEONIPIGA Trial: Neoadjuvant Nivolumab Plus Ipilimumab in MSI-H/dMMR Disease

The NEONIPIGA trial (NCT04006262) was a phase II study evaluating neoadjuvant nivolumab and ipilimumab, followed by surgery and adjuvant nivolumab, in patients with resectable dMMR/MSI-H gastric or GEJ AC [102]. Among 32 enrolled patients, 29 underwent surgery, all achieving R0 resection, with a remarkable pathological complete response (pCR) rate of 58.6%. Three patients who declined surgery had complete endoscopic responses. Grade 3/4 adverse events occurred in 19%, and surgical morbidity was 55%. With a median follow-up of 14.9 months, no recurrences were reported. The study supports dual immune checkpoint blockade as a promising chemotherapy-free neoadjuvant strategy for this molecularly defined subgroup.

#### 3.4.6. AI-Based Digital Score for Immunotherapy Prediction

An innovative study introduced an Artificial Intelligence (AI)-based digital score designed to predict patient benefit from maintenance immunotherapy in advanced GEJ AC [103]. Using pathology slide-derived features of the tumor microenvironment and spatial patterns, the model stratified likely responders, highlighting the shift toward AI-driven personalized immunotherapy.

Several emerging combinations are being explored to improve outcomes in advanced GEJ cancer. Key emerging targeted therapies under investigation are summarized in Table 3.

In the phase I REGONIVO trial, regorafenib plus nivolumab showed an ORR of 44% and median PFS of 5.6 months, with manageable safety and promising activity in the gastric cancer subgroup [104].

The EPOC1706 trial evaluated lenvatinib combined with pembrolizumab in first line and second-line settings, showing an impressive ORR of 69%, suggesting substantial antitumor efficacy [105].

Additionally, the INTEGRATE IIb trial (NCT0487936), an ongoing international phase III randomized study, is comparing regorafenib plus nivolumab to standard chemotherapy in patients with pretreated advanced GEJ cancer [106].

Collectively, these recent trials illustrate the rapid evolution of immunotherapy and targeted strategies in gastroesophageal cancer. Peshin et al., provided a comprehensive cross-tumor synthesis of immunotherapy advances across gastrointestinal malignancies, summarizing lessons from pivotal trials such as CheckMate 649, KEYNOTE 590, and ATTRACTION 3. Their analysis highlights shared mechanisms of response and resistance and underscores how biomarker-guided integration of immune checkpoint inhibitors is transforming standards of care throughout the GI oncology spectrum. These insights contextualize the therapeutic trends observed in metastatic GEC and set the stage for emerging molecularly targeted strategies discussed in the next section [107].

Despite substantial progress, resistance to systemic therapy remains a major barrier in metastatic GEC. Primary resistance to immunotherapy is often driven by non-inflamed tumor microenvironments, low tumor mutational burden, downregulation of antigen presentation pathways, and enrichment of immunosuppressive cell populations. Acquired resistance mechanisms include loss of HLA expression, upregulation of alternate immune checkpoints (such as TIM-3 and LAG-3), and dynamic remodeling of the tumor’s metabolic environment. Understanding these pathways is critical for designing rational combinations that overcome immune escape and improve treatment durability.

## 4. Emerging Therapeutics

Over the years, numerous efforts have been made to identify and evaluate novel therapeutic targets for GEJ cancers. Unfortunately, many of these clinical trials investigating targeted therapies, whether administered alone or in combination with chemotherapy, have yielded disappointing results. Notable examples include agents targeting various pathways, such as trastuzumab emtansine and lapatinib (*HER2*), cetuximab and panitumumab (*EGFR*), bevacizumab (*VEGF*), onartuzumab and rilotumumab (*MET*), everolimus (mTOR), and olaparib (PARP) [108,109,110,111,112,113,114,115,116,117].

Nonetheless, recent translational research had provided renewed momentum in this field. Advances in molecular profiling, immune-oncology, and biomarker-driven drug development have identified several novel and mechanistically distinct therapeutic targets that hold potential to redefine the treatment paradigm. The emerging targets and ongoing investigations are summarized in Table 4. The following sections outline (i) emerging translational and preclinical insights that are shaping new therapeutic hypothesis and (ii) late-phase and near-approval developments with immediate clinical relevance.

### 4.1. Translational Advances and Novel Preclinical Insights

Recent translational research has deepened understanding of tumor biology in gastroesophageal cancers, revealing pathways that mediate immune resistance, angiogenesis, and oncogenic signaling. Preclinical models have shown that activation of Wnt/ β-catenin and *DKK1* signaling contributes to immune exclusion and reduced responsiveness to immune checkpoint blockade, establishing *DKK1* as a promising immunomodulatory target [118]. Similarly, *FGFR2b* overexpression has been associated with epithelial–mesenchymal transition, angiogenesis, and immune suppression, providing the rationale for *FGFR2*-directed therapy [119]. Combination strategies integrating TKIs with immunotherapy (e.g., lenvatinib or regorafenib with PD-1 blockade) have demonstrated synergistic immune activation through a reduction in tumor-associated macrophages and increased CD8+ T-cell infiltration in preclinical and translational studies [120,121]. These insights are reshaping the design of next-generation clinical trials, underscoring the value of molecular profiling, ctDNA tracking, and artificial intelligence-based histopathologic modeling for predictive response assessment.

Future therapeutic innovation will require deeper interrogation of resistance biology. Preclinical models in GEC have shown that chronic MAPK and PI3K pathway activation, stromal remodeling, and epithelial-to-mesenchymal transition can drive resistance to *HER2*-, *FGFR2*-, and *CLDN18.2*-targeted therapies. Additionally, immunotherapy resistance is linked to myeloid-dominant microenvironments, exclusion of cytotoxic T cells, and Wnt/β-catenin activation. Therapeutic strategies targeting these mechanisms, including epigenetic modulators, stromal-directed agents, and novel checkpoint pathways, represent ongoing areas of translational development.

### 4.2. Near-Approval and Late-Phase Therapeutic Developments

Building on this translational foundation, several agents have now advanced into late-phase clinical testing or received regulatory approval:Claudin 18.2: Zolbetuximab (FDA 2024; SPOTLIGHT/GLOW) demonstrated significant PFS and OS benefit in HER2-negative, CLDN18.2-positive disease.FGFR2b: Bemarituzumab (FIGHT; FORTITUDE-101) showed encouraging trends toward improved survival.HER2: Zanidatamab (HERIZON-GEA-01) achieved an 84% response rate in phase 2, with phase 3 still ongoing.DKK1: DKN-01 (DisTinGuish) combined tislelizumab ± chemotherapy yielded 68–90% response rates in DKK1-high tumors.TKI-IO combinations: Lenvatinib + pembrolizumab (EPOC1706, LEAP-015) and regorafenib + nivolumab (REGONIVO, INTEGRATE IIb) are defining new chemo-free immune-targeted paradigms.

Together, these late-phase programs illustrate a shift from empirical cytotoxic therapy toward biomarker-defined, immune-integrated strategies, bridging preclinical mechanistic discoveries with translational clinical practice.

### 4.3. Claudin 18.2

Claudin 18 isoform 2 (*CLDN18.2*) is a tight junction protein predominantly expressed in normal gastric epithelial cells but is aberrantly activated in several malignancies, including those of the stomach, pancreas, esophagus, ovary, and lung. Zolbetuximab, a monoclonal antibody that selectively targets *CLDN18.2*, demonstrated clinical benefit in the phase 2 FAST trial. In this study, combining zolbetuximab with chemotherapy (EOX: epirubicin, oxaliplatin, and capecitabine) led to a notable improvement in survival outcomes for patients with advanced *CLDN18.2*-positive gastric and GEJ AC [122].

Building on these results, the phase 3 SPOTLIGHT trial presented at the 2023 ASCO Gastrointestinal Cancers Symposium assessed zolbetuximab combined with FOLFOX (5-fluorouracil and oxaliplatin) in a first-line setting. In *HER2*-negative, *CLDN18.2*-positive advanced gastric/GEJ AC, this combination significantly extended PFS (10.6 months vs. 8.7 months; HR 0.75; *p* = 0.0066) and OS (18.2 months vs. 15.5 months; HR: 0.75; *p* = 0.0053), compared to chemotherapy alone [93].

Another pivotal study, the GLOW trial, evaluated zolbetuximab with CAPOX (capecitabine and oxaliplatin). This trial met both its primary endpoint (PFS) and secondary endpoint (OS) in a similar population of *CLDN18.2*-positive, *HER2*-negative patients, further supporting the clinical utility of this targeted approach [123]. Collectively, these findings highlight *CLDN18.2* as a compelling predictive biomarker for targeted therapy. The favorable results from both the SPOTLIGHT and GLOW trials suggest that zolbetuximab combined with chemotherapy may represent a new standard of care for patients within this molecularly defined subgroup. Notably, only 13.2% of *CLDN18.2*-positive patients in the SPOTLIGHT trial exhibited a PD-L1 combined positive score (CPS) ≥ 5, and all enrolled participants were *HER2*-negative. This underscores a previously underserved population that had limited access to effective targeted therapies. Ongoing research is actively investigating next-generation approaches directed at *CLDN18.2*. These include monoclonal antibodies, antibody-drug conjugates, bispecific antibodies, and cell-based immunotherapies (e.g., NCT04856150, NCT04400383, NCT04632108, NCT04805307, NCT04495296, NCT04404595). Furthermore, zolbetuximab is also being tested in combination with ICIs (e.g., NCT03505320), with the goal of achieving enhanced anti-tumor activity through synergistic immune modulation. The evolution of systemic therapy in advanced gastroesophageal cancer across major clinical milestones is illustrated in Figure 4.

### 4.4. FGFR2

Fibroblast growth factor receptors (FGFRs) have gained attention as therapeutic targets in various cancers, with *FGFR2* emerging as particularly promising in gastroesophageal malignancies [119,120]. Bemarituzumab, a monoclonal antibody selectively targeting the *FGFR2b* isoform, has demonstrated potential in this setting based on both preclinical and clinical studies [124].

The FIGHT trial, a phase 2 randomized, double-blind, placebo-controlled study, evaluated the efficacy of bemarituzumab in combination with FOLFOX (5-fluorouracil, leucovorin, and oxaliplatin) in patients with *FGFR2b* overexpression or gene amplification. Participants received either bemarituzumab + FOLFOX (*n* = 77) or placebo + FOLFOX (*n* = 78). Although the trial did not meet its primary endpoint of PFS, the results showed encouraging trends. The median PFS was 9.5 months in the treatment arm vs. 7.4 months in the control (HR 0.68; *p* = 0.073). Additionally, OS was longer in the bemarituzumab group (not reached vs. 12.9 months, HR 0.58; *p* = 0.027), and the objective response rate (ORR) was higher (47% vs. 33%) [75].

The trial also revealed that nearly 30% of patients with *HER2*-negative advanced gastric or GEJ AC exhibited *FGFR2b* overexpression or gene amplification, highlighting a substantial subgroup that may benefit from *FGFR2*-directed therapies.

To confirm these findings, the phase 3 FORTITUDE-101 trial is currently in progress, investigating bemarituzumab combined with FOLFOX as a first-line treatment in *FGFR2b*-selected patients with advanced gastric or GEJ AC (NCT05052801).

The encouraging outcomes from *FGFR2b*-selective therapies must be interpreted cautiously due to small *FGFR2b*-positive sample sizes and significant geographic concentration in Asia. In addition, assays for *FGFR2b* overexpression differ among studies, complicating reproducibility. The absence of phase III survival data limits the ability to establish *FGFR2b*-targeted treatments as definitive first-line standards.

ctDNA is emerging as a powerful biomarker for real-time treatment monitoring in gastroesophageal cancers. Early decreases in ctDNA correlate with treatment response, while rising ctDNA levels can detect recurrence or therapeutic resistance months earlier than imaging. Integrating ctDNA surveillance into clinical trials of *CLDN18.2*- and *FGFR2*-targeted therapies may enable adaptive therapy strategies, earlier recognition of resistance, and personalized escalation or de-escalation of treatment [45,46,47].

### 4.5. DKK1

The canonical Wnt/β-catenin signaling pathway plays a crucial role in cellular proliferation, migration, and survival, with its dysregulation contributing to the pathogenesis of various malignancies, including gastric and gastroesophageal junction (GEJ) cancers [125,126,127]. While Dickkopf-1 (*DKK1*) is classically recognized as a Wnt signaling antagonist, emerging evidence highlights its paradoxical role in promoting tumor progression, immune evasion, and angiogenesis [128,129,130].

Increased *DKK1* expression has been linked to a more immunosuppressive tumor microenvironment (TME), which hampers immune surveillance and undermines the efficacy of ICIs. In this context, DKN-01, a humanized monoclonal antibody that neutralizes *DKK1*, has been developed as a promising agent to restore immune responsiveness.

DKN-01 is currently being evaluated in the phase 2 DisTinGuish trial (NCT04363801), where it is administered alongside tislelizumab (anti–PD-1) and chemotherapy (FOLFOX or CAPOX) in patients with *HER2*-negative advanced gastric or GEJ AC. Interim results have demonstrated encouraging clinical activity: among 25 patients treated in the first-line setting, the ORR was 68%, with an even more remarkable 90% ORR in patients with high *DKK1* expression, suggesting *DKK1* may serve as a predictive biomarker [131,132].

A 2025 study by Shi and Wei, published in the Journal of Clinical Oncology, further reinforces the therapeutic rationale. The study demonstrated that inhibiting *DKK1* remodeled the tumor microenvironment (TME), facilitating immune cell infiltration and enhancing responsiveness to ICIs, especially in immunologically “cold” tumors [118].

Together, these findings underscore the dual role of *DKK1* as both a therapeutic target and a biomarker. The ongoing DisTinGuish trial, along with future randomized studies, will be crucial in validating this approach and defining its role in the personalized treatment of gastric and GEJ cancers.

Despite promising early results, *DKK1*-directed therapies face limitations, including the small number of *DKK1*-high patients and lack of randomized controlled data. Moreover, *DKK1* assays are not standardized across laboratories, which may lead to misclassification of patients. Its future utility will depend on harmonization of testing and validation in larger, more diverse populations.

### 4.6. Tyrosine Kinase Inhibitors (TKIs)

The combination of multi-kinase inhibitors (MKIs) with ICIs has become a transformative strategy in various cancers, including renal cell carcinoma and endometrial cancer. This approach is now gaining traction in GEJ cancers, where oncogenic tyrosine kinase signaling not only drives tumor growth but also promotes an immunosuppressive TME [133]. Preclinical models have shown that TKI-mediated inhibition of these pathways can lead to a reduction in tumor-associated macrophages and a corresponding increase in CD8⁺ cytotoxic T-cell infiltration, ultimately converting “cold” tumors into immune-responsive microenvironments [121,134].

One of the most promising TKI-IO combinations in GEJ cancer is lenvatinib plus pembrolizumab, investigated in the EPOC1706 phase 2 trial involving 29 patients with advanced gastric AC. The study reported an impressive objective response rate (ORR) of 69%, highlighting the potential synergy of this combination [105].

To further validate the promising activity of tyrosine kinase inhibitor (TKI) and immunotherapy combinations, two pivotal phase 3 trials are currently underway. LEAP-014 (NCT04949256) is evaluating the combination of lenvatinib, pembrolizumab, and chemotherapy as a first-line treatment for patients with advanced esophageal squamous cell carcinoma (ESCC). In parallel, LEAP-015 (NCT04662710) is investigating the same regimen in patients with GEJ AC, also in the first-line setting.

Beyond lenvatinib, several other TKI-based regimens are under clinical evaluation. Cabozantinib combined with atezolizumab is being tested as a second-line therapy for recurrent or metastatic ESCC in an ongoing phase 2 trial (NCT05007613). Another investigational combination, sitravatinib plus tislelizumab, is in development for heavily pretreated ESCC patients, with evaluation currently ongoing in NCT05461794.

Emerging research between 2023 and 2025 continues to expand the therapeutic landscape. A 2025 preclinical study demonstrated that the combination of regorafenib and anti–PD-1 therapy triggered robust immune activation in GEJ cancer models, paving the way for anticipated early-phase clinical testing. Additionally, a pilot study scheduled in Korea for 2024 will assess fruquintinib, a *VEGFR* inhibitor, in combination with camrelizumab for patients with *VEGFR*-high gastroesophageal cancers.

Further experimental efforts include early-stage trials involving axitinib, vandetanib, and apatinib, each paired with PD-1 inhibitors. These studies are designed to explore novel immune-TKI synergies, with initial data expected in 2025, marking an exciting evolution in the development of multi-targeted immunomodulatory strategies for gastroesophageal cancers.

Several clinical trials are now assessing the synergistic potential of combining tyrosine kinase inhibitors (TKIs) with immune checkpoint blockade in gastroesophageal cancer, as outlined in Table 5.

Artificial intelligence and machine-learning models are increasingly utilized to characterize the tumor-immune microenvironment, quantify spatial relationships, and identify predictive signatures for therapy selection. Digital pathology-based scoring systems and multimodal AI platforms that integrate histology, radiomics, and transcriptomics may allow more accurate stratification for immunotherapy, TKI-IO combinations, and novel targeted agents. Early studies demonstrate that AI-derived TME scores outperform conventional biomarkers such as PD-L1 CPS, supporting their future integration into clinical trial design and personalized treatment algorithms.

TKI-IO combination studies differ widely in design, dosing, and patient selection, limiting cross-study comparison. Many early phase studies lacked control arms, and most enrolled primarily in East Asian patients. Differences in *VEGFR* expression prevalence and TKI toxicity profiles across populations further complicate interpretation. Ongoing phase III trials will be critical to determine whether these combinations offer consistent benefit across global settings.

### 4.7. Cell-Based Immunotherapies: Emerging Role of CAR T-Cells

Chimeric antigen receptor (CAR) T-cell therapy has revolutionized the management of hematologic malignancies, achieving durable remissions in otherwise refractory disease (Figure 5). Translating this success to solid tumors such as gastric and GEJ cancers has proven challenging due to a lack of truly tumor-specific antigens, the immunosuppressive tumor microenvironment, and the risk of severe immune-mediated toxicities [135,136,137]. Recent innovations in CAR design, including dual-antigen targeting, switchable safety domains, and improved trafficking, are gradually overcoming these barriers and renewing optimism about the applicability of cell-based immunotherapy in gastrointestinal oncology [135]. Similar translational barriers have been described in pancreatic cancer, where CAR T-cell activity is also constrained by stromal density, antigen heterogeneity, and tumor microenvironment-mediated immunosuppression, highlighting shared challenges across gastrointestinal malignancies [136].

One of the most promising developments is Claudin-18.2 (*CLDN18.2*)-directed CAR T-cell therapy. Satricabtagene autoleucel (satri-cel), an autologous *CLDN18.2*-specific CAR T product, has demonstrated clinically meaningful activity in heavily pretreated gastric and GEJ cancers. In a phase 2 randomized trial (NCT04581473), satri-cel significantly prolonged median progression-free survival compared with standard therapies (3.25 vs. 1.77 months; HR 0.37, *p* < 0.0001), with a manageable safety profile characterized mainly by cytopenias and cytokine-release syndrome [135,138,139].

Beyond *CLDN18.2*, multiple other targets including *HER2*, *EGFR*, *CEA*, *mesothelin* (*MSLN*), *CD276*, and *NKG2D*, are under investigation in preclinical and early phase trials. These studies have reported encouraging tumor regression and immune persistence, particularly with *HER2*- and *CLDN18.2*-directed constructs [135,137,138]. The majority of ongoing trials, primarily conducted in East Asia, are assessing safety, cytokine toxicity mitigation, and long-term durability of responses. Although CAR T-cell therapy in solid tumors remains in its early stages, these developments underscore its growing translational potential and position it as an emerging pillar in the therapeutic armamentarium for advanced gastroesophageal malignancies [135,137,140].

While CAR T-cell therapies are emerging as a potential therapeutic strategy, interpretation of early trials requires caution. Most studies have very small cohorts, lack control arms, and employ heterogenous CAR constructs. Geographic restriction to East Asian populations further limits generalizability. Manufacturing variability, antigen-loss relapse, and cytokine toxicity remain major barriers to widespread adoption.

Looking ahead, the next phase of GEC research will focus on integrating mechanistic insights with precision-guided treatment. Strategies aimed at overcoming immune resistance, targeting stromal and metabolic determinants of tumor progression, and leveraging ctDNA and AI-driven biomarker platforms promise to refine patient selection and enhance therapeutic efficacy. Continued harmonization of biomarker assays and incorporation of dynamic monitoring tools will be essential to translate emerging scientific discoveries into durable clinical benefit.

### 4.8. Nanoscale Imaging and Diagnostic Technologies in GEC

In parallel with advances in molecularly targeted therapies, disruptive nanoscale imaging technologies are emerging as powerful tools for early cancer detection and treatment monitoring. Ultrahigh-resolution platforms such as scattering-type scanning near-field optical microscopy (s-SNOM) and other vibrational nano-spectroscopic techniques enable label-free visualization of biomolecular architecture at the nanoscale, providing unprecedented insights into tumor heterogeneity and microenvironmental remodeling [141]. Recent work has shown that nanoscale morphodynamic signatures in esophageal cancer cells, including alterations in membrane roughness, stiffness, and cytoskeletal organization, can function as quantitative fingerprints of drug response, allowing investigators to assess therapeutic efficacy at single-cell resolution [142]. Incorporation of these nanoscale diagnostic approaches may complement existing biomarker-driven strategies and facilitate earlier, more accurate evaluation of treatment benefit in gastroesophageal cancers.

It is important to note that demographic characteristics, including sex, ethnicity, and age, were not uniformly balanced across the included trials. Several East Asian studies had limited non-Asian participation, while global trials often had male-predominant enrollment. Moreover, older adults were underrepresented in many phase II-III investigations. These demographic imbalances may introduce bias and affect the external validity of trial outcomes. Although our review relies on published aggregate data and cannot adjust for these variables, we underscore the need for future studies to ensure more diverse representation to improve applicability across patient populations.

### 4.9. Limitations

This narrative review has several limitations. First, the evidence base for gastroesophageal cancers is heterogenous, with substantial variation in trial designs, geographic representation, and biomarker assay methodologies, which complicates cross-study comparisons. Many emerging therapies have been evaluated in small or biomarker-restricted subgroups, limiting generalizability, while some data, particularly for MSI-H/dMMR disease, derive from retrospective analysis or exploratory cohorts. Demographic imbalances, including underrepresentation of older adults and regional differences in sex and ethnicity distributions, further constrain external validity. Additionally, the absence of direct head-to-head comparisons restricts definitive ranking of therapeutic strategies. These limitations should be considered when interpreting the findings summarized in this review.

## 5. Conclusions

Across the therapeutic landscape, a direct comparison of treatment modalities is challenged by heterogenous trial designs, inconsistent biomarker assays, and significant geographic variation in patient populations. These sources of variability complicate the interpretation of outcomes and limit the generalizability of trial results to global practice. Future studies must prioritize harmonized biomarker testing, standardized endpoints, and diverse enrollment to enable more accurate cross-trial comparisons. The therapeutic landscape of gastroesophageal cancer has undergone a remarkable transformation over the past decade, driven by advances in molecular profiling, immuno-oncology, and biomarker-directed strategies. Treatment is no longer a one-size-fits-all approach but rather a nuanced, multidisciplinary algorithm tailored to tumor biology, disease stage, and patient fitness. For localized disease, perioperative chemotherapy—now evolving to incorporate immunotherapy—is redefining curative strategies. In advanced settings, the integration of *HER2*-targeted agents, immune checkpoint inhibitors, and emerging molecular targets has improved outcomes in biomarker-selected populations. However, challenges remain, including resistance mechanisms, the limited efficacy of therapies in unselected patients, and access to next-generation agents. Ongoing trials exploring novel combinations, bispecific antibodies, and tumor microenvironment modulation hold promise for the next wave of innovation. As precision oncology continues to advance, translating molecular insights into durable, real-world clinical benefit will be key to improving survival and quality of life for patients with this aggressive malignancy.

## Figures and Tables

**Figure 1 ijms-26-11424-f001:**
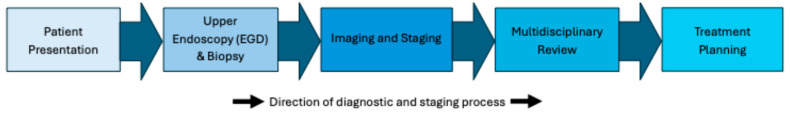
Simplified diagnostic and staging pathway for localized gastroesophageal cancer. This figure outlines the sequential steps in the diagnostic and staging process for localized gastroesophageal cancer, beginning with clinical presentation and proceeding through endoscopy with biopsy, imaging, and staging evaluations, multidisciplinary tumor board review, and treatment planning. Arrows indicate the recommended direction of workflow. Biomarker testing for *HER2*, PD-L1, and *MSI-H/dMMR* should be performed where indicated, and staging laparoscopy is recommended for gastric or gastroesophageal junction tumors.

**Figure 2 ijms-26-11424-f002:**
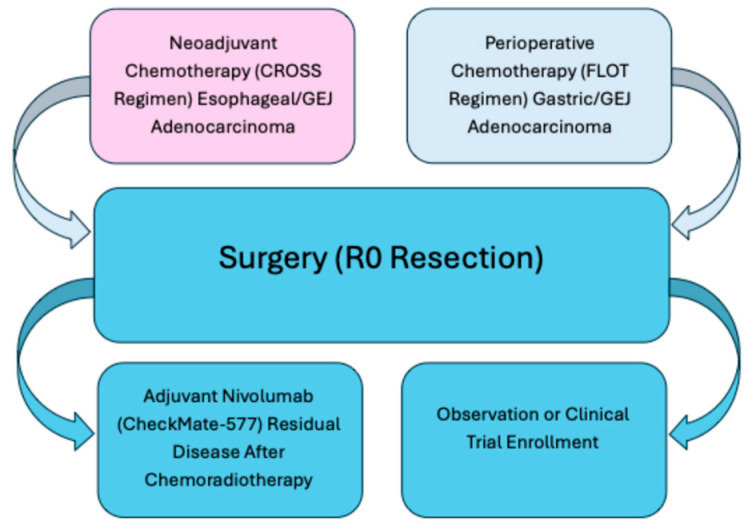
Treatment pathways for localized gastroesophageal cancer. This figure summarizes standard curative-intent treatment approaches for localized gastroesophageal adenocarcinoma. Neoadjuvant chemoradiotherapy (CROSS regimen) is preferred for esophageal and gastroesophageal junction tumors, whereas perioperative chemotherapy (FLOT regimen) is standard for gastric and select GEJ cancers. Both strategies converge at surgical resection (R0). Following surgery, patients with residual disease after chemoradiotherapy may benefit from adjuvant nivolumab (CheckMate-577), while observation or clinical trial enrollment may be appropriate for others. FLOT = 5-fluorouracil, leucovorin, oxaliplatin, docetaxel.

**Figure 3 ijms-26-11424-f003:**
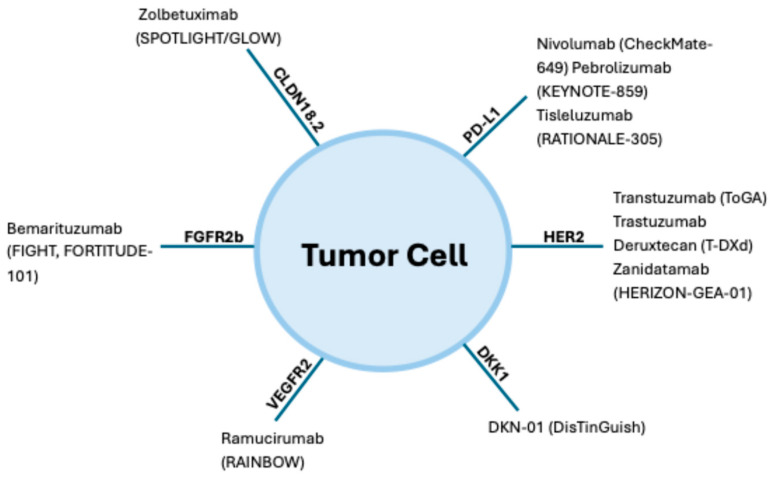
Biomarker landscape and therapeutic targets in gastroesophageal cancer. This figure summarizes key approved and investigational agents (with pivotal trials): HER2 (trastuzumab (CheckMate-649), pembrolizumab (KEYNOTE-859), tislelizumab (RATIONALE-305)); CLDN18.2 (zolbetuximab (SPOTLIGHT/GLOW)); FGFR2b (bemarituzumab (FIGHT, FORTITUDE-101)); VEGFR2 (ramucirumab (RAINBOW)); DKK1 (DKN-01 (DisTinGuish)).

**Figure 4 ijms-26-11424-f004:**
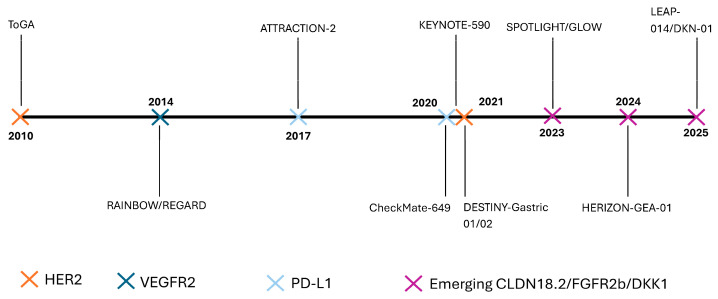
Evolution of systemic therapy in advanced gastroesophageal cancer (2010–2025). This figure summarizes key milestones shaping systemic therapy for advanced gastroesophageal cancer. The timeline illustrates the progression from the *HER2*-targeted ToGA trial (2010) through the establishment of anti-angiogenic (RAINBOW/REGARD) and immune checkpoint blockade (ATTRACTION-2, CheckMate 649, KEYNOTE-590) regimens, to the introduction of emerging biomarker-directed agents targeting *CLDN18.2*, *FGFR2b*, and *DKK1* (SPOTLIGHT, HERIZON-GEA-01, LEAP-014). Together, these milestones highlight the transformation from empiric chemotherapy to precision-guided and immune-based treatment paradigms.

**Figure 5 ijms-26-11424-f005:**
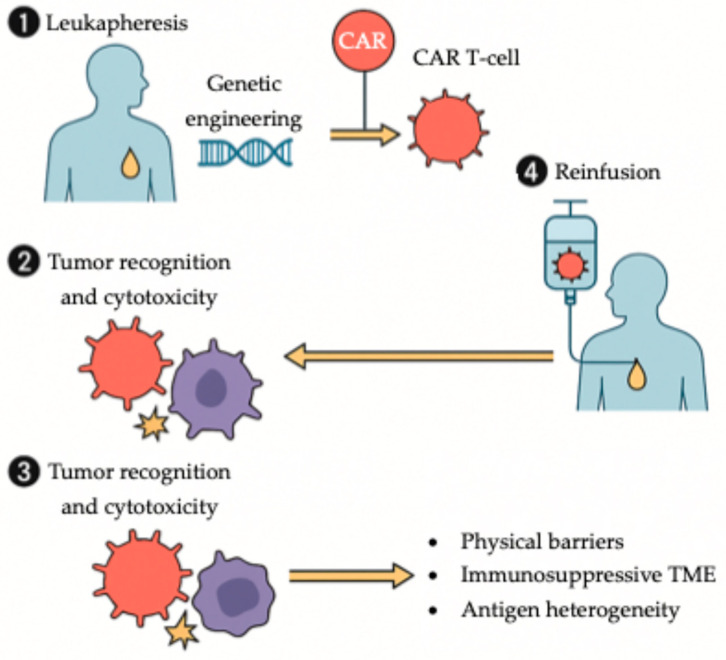
Workflow of CAR T-cell therapy in solid tumors. Autologous T cells are collected via leukapheresis and genetically engineered ex vivo to express a chimeric antigen receptor (CAR). Following expansion, CAR T cells are reinfused into the patient, where they recognize and eliminate tumor cells. In gastroesophageal cancers, therapeutic efficacy is modulated by several challenges, including physical barriers, an immunosuppressive tumor microenvironment, and tumor antigen heterogeneity.

**Table 1 ijms-26-11424-t001:** Selected Ongoing Trials Investigating Immunotherapy in Localized GEC.

Trial Name	Design & Population	Intervention	Control	Key Outcomes	Potential Side Effects
DANTE (NCT03421288)	Phase II, perioperative setting, resectable gastric or GEJ adenocarcinoma	FLOT + Atezolizumab (anti–PD-L1)	FLOT alone	mOS 23.2 mo	Neutropenia, diarrhea, neuropathy, immune related side effects (rash, thyroiditis, hepatitis)
VESTIGE (NCT03443856)	Phase II, adjuvant setting, residual disease after neoadjuvant chemo and resection	Adjuvant Nivolumab ± Ipilimumab	Adjuvant chemo (FOLFOX/CAPOX)	mDFS 22.4 vs. 11.0 mo	Immune related side effects (colitis, dermatitis, endocrinopathies), fatigue, cytopenias
NeoFLOT (NCT02872103)	Phase III, neoadjuvant setting, resectable *HER2*-negative gastric or GEJ adenocarcinoma	FLOT + Nivolumab	Historical FLOT data	mOS 50 mo	Neutropenia, GI toxicity, immune related side effects (fatigue, rash, thyroid dysfunction)
MATTERHORN (NCT045929213)	Phase III, perioperative setting, resectable gastric or GEJ adenocarcinoma	Durvalumab + chemotherapy vs. chemotherapy alone	Resectable gastric/GEJ adenocarcinoma	mOS 47.2 mo; pCR 11.5% vs. 7.1% (*p* < 0.01)	Neutropenia, nausea, immune related side effects (pneumonitis, hepatitis, endocrinopathy)

**Table 2 ijms-26-11424-t002:** Major Clinical Trials in Metastatic/Unresectable Gastroesophageal Cancer.

Trial Name	Target/Focus	Setting (Line)	Intervention	Key Outcomes	Potential Side Effects	Status
SPOTLIGHT (NCT03504397)	*CLDN18.2* + GC/GEJ	First-line	Zolbetuximab + mFOLFOX6 vs. placebo	PFS 10.6 vs. 8.7 months; HR 0.75; *p* = 0.0066; OS 18.2 vs. 15.5 months; HR 0.75; *p* = 0.0053	Nausea, vomiting, diarrhea, neutropenia, anemia, fatigue	Active, not recruiting
GLOW (NCT03462719)	*CLDN18.2* + GC/GEJ	First-line	Zolbetuximab + CAPOX vs. placebo	PFS 8.2 vs. 6.8 months; HR 0.687; *p* = 0.0007; 18.2 vs. 15.5 months; HR 0.75; *p* = 0.0053	Nausea, vomiting, diarrhea, neutropenia, peripheral edema	Active, not recruiting
HERIZON-GEA-01 (NCT05152147)	*HER2* + GC/GEJ	First-line	Zanidatamab + chemo	ORR 84%; mPFS 15.2 mo	Diarrhea, infusion-related reactions, fatigue, cytopenia	Active, not recruiting
KEYNOTE-859 (NCT03675737)	*HER2*-negative GC/GEJ	First-line	Pembrolizumab + chemo vs. chemo	OS 12.9 vs. 11.5 mo; HR 0.78 (95% CI 0.70–0.87); PFS 6.9 vs. 5.6 mo; HR 0.76 (95% CI 0.67–0.85)	Neutropenia, anemia, hypothyroidism, pruritis, hepatitis, colitis, pneumonitis	Completed
LEAP-014 (NCT04949256)	ESCC	First-line	Pembrolizumab + lenvatinib + chemo	ORR 86%; mDoR 11.7 mo	Hypertension, diarrhea, fatigue, proteinuria, hand-foot syndrome, thyroid dysfunction, rash, hepatitis	Active, not recruiting
DESTINY-Gastric01 (NCT03329690)	*HER2* + GC/GEJ	Third line	Trastuzumab deruxtecan vs. chemo	OS 12.5 vs. 8.4 mo	Neutropenia, anemia, nausea, vomiting, fatigue, interstitial lung disease, pneumonitis	Completed
INTEGRATE IIa (NCT02773524)	Advanced AGOC	Third line and beyond	Regorafenib vs. placebo	OS 4.5 vs. 4.0 mo; HR 0.52; *p* = 0.011	Hand-foot skin reaction, hypertension, asthenia, diarrhea, hoarseness	Unknown status
INTEGRATE IIb (NCT04879368)	Advanced AGOC	Beyond second line	Regorafenib + nivolumab vs. chemo	Ongoing	Cytopenia, hypertension, rash, dry skin, fatigue, diarrhea, immune-related side effects (thyroid dysfunction, hepatitis, colitis)	Active, not recruiting
FRUTIGA (NCT03223376)	Advanced GC/GEJ	Second-line	Fruquintinib + paclitaxel vs. placebo	PFS 5.6 vs. 2.7 mo; HR 0.57; *p* < 0.0001; OS 9.6 vs. 8.4 months; HR 0.96; *p* = 0.6064; ORR 42.5% vs. 22.4%; DCR 77.2% vs. 56.3%	Neutropenia, leukopenia, anemia, hypertension, proteinuria, fatigue	Unknown status
REGONIVO (NCT03406871)	Advanced GC	Beyond second line	Regorafenib + nivolumab	ORR 44%; PFS 5.6 mo	Fatigue, hypertension, diarrhea, rash, skin toxicity, immune-related side effects (thyroid dysfunction, hepatitis, colitis)	Completed
EPOC1706 (NCT03609359)	Advanced GC	First-/Second line	Lenvatinib + pembrolizumab	ORR 69%	Hypertension, diarrhea, fatigue, proteinuria, immune-related side effects (hypothyroidism, rash, hepatitis)	Completed
ATTRACTION-2 (NCT02267343)	Advanced GC/GEJ	Third line	Nivolumab vs. placebo	OS 5.3 vs. 4.1 mo; HR 0.63 (95% CI 0.51–0.78, *p* < 0.0001)	Fatigue, diarrhea, proteinuria, immune-related side effects (hypothyroidism, pneumonitis, hepatitis, colitis)	Completed
ATTRACTION-4 (NCT02746796)	Advanced GC/GEJ	First line	Nivolumab + SOX or CAPOX vs. Chemotherapy alone or + placebo	PFS 10.45 vs. 8.03 mo; HR 0.68 (98.51% CI 0.51–0.90)	Neutropenia, thrombocytopenia, anemia, immune-related side effect (thyroid dysfunction, colitis, hepatitis, pneumonitis)	Completed
KEYNOTE-059 (NCT02335411)	Advanced GC	Third line	Pembrolizumab	ORR 11.6% vs. 6.4%; mDoR 8.4 vs. 16.3 vs. 6.9 mo	Fatigue, nausea, diarrhea, pruritis, immune-related side effects (hypothyroidism, pneumonitis, hepatitis, colitis)	Completed
RAMIRIS (NCT03081143)	Advanced GC (post-FLOT)	Second-line	FOLFIRI + ramucirumab vs. paclitaxel + ramucirumab	Higher ORR (25% vs. 8%)	Neutropenia, leukopenia, diarrhea, asthenia, hypertension, proteinuria, mucositis	Active, not recruiting
RAINBOW (NCT01170663)	Advanced GC/GEJ	Second-line	Ramucirumab + paclitaxel vs. placebo + paclitaxel	OS: 9.6 vs. 7.4 mo; HR 0.807, *p* = 0.017	Neutropenia, leukopenia, febrile neutropenia, asthenia, hypertension, diarrhea, peripheral edema	Completed

**Table 3 ijms-26-11424-t003:** Emerging Targeted Therapies in Metastatic GECs.

Target	Therapy	Trial	Setting	Key Results	Potential Side Effects	Status
*CLDN18.2*	Zolbetuximab	SPOTLIGHT (NCT03504397), GLOW (NCT03653507)	First-line	OS 18.2 vs. 15.5 months; PFS 10.6 vs. 8.7 months	Nausea, vomiting, febrile neutropenia, thrombocytopenia	Active, not recruiting
*FGFR2b*	Bemarituzumab	FIGHT (NCT03694522)	First-line *FGFR2b*+	Median PFS 9.5 vs. 7.4 months; median OS 19.2 vs. 13.5 months	Neutropenia, anemia, GI side effects, infusion reaction	Completed
*HER2* (Bispecific)	Zanidatamab	HERIZON-GEA-01 (NCT05152147)	First-line	ORR 84%; mPFS 15.2 months	Diarrhea, infusion-related infections, cytopenia	Active, not recruiting
*VEGFR2*	Ramucirumab	RAINBOW (NCT01170663)	Second-line	OS 9.6 vs. 7.4 months	Febrile neutropenia, hypertension, fatigue, peripheral edema	Completed
*VEGFR2*	Fruquintinib	FRUTIGA (NCT03223376)	Second-line	Median PFS 5.6 vs. 2.7 months; median OS 9.6 vs. 8.4 months; ORR 42.5% vs. 22.4%	Hypertension, palmar-plantar erythrodysesthesia, proteinuria, dysphonia, abdominal pain, diarrhea, asthenia	Unknown status
Multikinase + PD-1	Regorafenib + Nivolumab	REGONIVO (NCT03406871), INTEGRATE IIb (NCT04879368)	Later-line	ORR 44%; PFS 5.6 months	Fatigue, hypertension, hand-foot skin reaction, asthenia, immune-related side effects (hypothyroidism, colitis)	Completed; Active, not recruiting
PD-1 + *VEGFR* Inhibitor	Lenvatinib + Pembrolizumab	EPOC1706 (NCT03609359)	First/Second line	ORR 69%	Hypertension, diarrhea, fatigue, proteinuria, immune-related side effects	Completed

**Table 4 ijms-26-11424-t004:** Emerging Molecular Targets and Therapeutic Strategies in GEC.

Study Title	Phase	Intervention	Line	Condition	NCT ID	Target	Key Outcomes	Potential Side Effects
SPOTLIGHT	3	FOLFOX + Zolbetuximab	1L	Gastric/GEJ AC	NCT03504397	Claudin 18.2	OS 18.2 vs. 15.5 months; PFS 10.6 vs. 8.7 months	Nausea, vomiting, diarrhea, neutropenia, anemia, fatigue
GLOW	3	CAPOX + Zolbetuximab	1L	Gastric/GEJ AC	NCT03653507	Claudin 18.2	OS 14.4 vs. 12.2 months; PFS 8.2 vs. 6.8 months	Nausea, vomiting, diarrhea, neutropenia, peripheral edema
ILUSTRO	2	Zolbetuximab ± FOLFOX ± pembrolizumab/nivolumab	1L/3L+	Gastric/GEJ AC	NCT03505320	Claudin 18.2	ORR 71.4%; median PFS 17.8 months	Nausea, vomiting, abdominal pain, infusion reaction
FIGHT	2	FOLFOX + Bemarituzumab	1L	Gastric/GEJ AC	NCT03694522	*FGFR2b*	Median PFS 9.5 vs. 7.4 months; median OS 19.2 vs. 13.5 months	Neutropenia, anemia, GI side effects, infusion reaction
FORTITUDE-101	3	FOLFOX + Bemarituzumab	1L	Gastric/GEJ AC	NCT05052801	*FGFR2b*	Median OS 17.9 vs. 12.5 months; median PFS 8.6 vs. 6.7 months	Ocular surface toxicity, stomatitis, diarrhea, neutropenia
DisTinGuish	2	FOLFOX/CAPOX + Tislelizumab + DKN-01	1L/2L	Gastric/GEJ AC	NCT04363801	*DKK1*	Median OS 8.2 months; median PFS 1.4 months	Anemia, thrombocytopenia, fatigue, diarrhea, nausea
LEAP-014	3	Pembrolizumab + Lenvatinib + Chemotherapy	1L	Esophageal SCC	NCT04949256	TKI	Median OS 17.6 vs. 15.5 months; median PFS 7.2 vs. 6.9 months	Hypertension, diarrhea, fatigue, proteinuria, hand-foot syndrome, thyroid dysfunction, rash, hepatitis
LEAP-015	3	Pembrolizumab + Lenvatinib + Chemotherapy	1L	Gastroesophageal AC	NCT04662710	TKI	Median OS 12.6 vs. 12.9 months	Hypertension, diarrhea, hypothyroidism, fatigue, stomatitis, proteinuria
Cabozantinib + Atezolizumab	2	Atezolizumab + Cabozantinib	2L	Esophageal SCC	NCT05007613	TKI	Not reported	Diarrhea, fatigue, hypertension, palmar-plantar erythrodysesthesia, stomatitis
Sitravatinib + Tislelizumab	2	Tislelizumab + Sitravatinib	2L+	Esophageal SCC	NCT05461794	TKI	ORR 16%	Hypertension, diarrhea, fatigue, nausea, liver enzyme elevation

**Table 5 ijms-26-11424-t005:** Key Trials of TKIs + Immunotherapy in Gastroesophageal Cancers.

Trial Name	Phase	Intervention	Cancer Type	Treatment Line	NCT ID	Key Outcomes	Potential Side Effects
EPOC1706	Phase 2	Lenvatinib + Pembrolizumab	Advanced Gastric Cancer	1L/2L	NCT03609359	ORR 69%	Hypertension, diarrhea, fatigue, proteinuria, immune-related side effects
LEAP-014	Phase 3	Lenvatinib + Pembrolizumab + Chemotherapy	Esophageal SCC (ESCC)	1L	NCT04949256	Median OS 17.6 vs. 15.5 months; median PFS 7.2 vs. 6.9 months	Hypertension, diarrhea, fatigue, proteinuria, hand-foot syndrome, thyroid dysfunction, rash, hepatitis
LEAP-015	Phase 3	Lenvatinib + Pembrolizumab + Chemotherapy	Gastroesophageal Adenocarcinoma (GEJ AC)	1L	NCT04662710	Median OS 12.6 vs. 12.9 months	Hypertension, diarrhea, hypothyroidism, fatigue, stomatitis, proteinuria
—	Phase 2	Cabozantinib + Atezolizumab	Recurrent/Metastatic ESCC	2L	NCT05007613	Not reported	Hypertension, diarrhea, fatigue, hand-foot skin reaction, increased liver enzymes, immune-related side effects (thyroid dysfunction, colitis, hepatitis)
—	Phase 2	Sitravatinib + Tislelizumab	Heavily Pre-treated ESCC	2L+	NCT05461794	Not reported	Liver enzyme increase, anemia, hypoalbuminemia, hand-foot skin reaction, thrombocytopenia, hypertension
—	Preclinical	Regorafenib + Anti–PD-1	GEJ Models	—	—	Expected to enter early-phase testing (2025)	—
—	Planned (2024)	Fruquintinib + Camrelizumab	Gastroesophageal Cancers (*VEGFR*-high)	—	—	Korea-based pilot study	—
—	Early-stage	Axitinib, Vandetanib, Apatinib + PD-1 Inhibitors	Various Gastroesophageal Cancers	—	—	Trials anticipated by 2025	—

## Data Availability

No new data were created or analyzed in this study. Data sharing is not applicable to this article.

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
