# Peer review of "Therapeutic Frontiers in Gastroesophageal Cancer: Contemporary Concepts in Management and Therapy"

_ijms, 2025, doi:10.3390/ijms262311424_

Round 1
Reviewer 1 Report
Comments and Suggestions for Authors
Dear Authors,
First of all, congratulations for your interesting work. I hope that my hints will help you in the next steps of improvement and the final manuscript will be really valuable for the readers. Your review provides a comprehensive, up-to-date synthesis of diagnostic and therapeutic advances in gastroesophageal cancer. It integrates molecular profiling, immunotherapy, and targeted treatment data, contextualizing ongoing clinical trials. Its scientific value lies in consolidating current evidence and emphasizing biomarker-driven, precision-based management strategies in both localized and metastatic disease.
There are several punctuation mistakes (such as double space, double dot or no at all) and some typos - even if they do not change the value of the manuscript, I'd like to urge you to correct these imperfections. Some sentences are overly long and could be simplified for readability.
Something that requires attention is limited critical evaluation: the review mainly summarizes studies but rarely critiques their limitations or conflicting evidence. Include comparative analysis of trial designs, geographic biases, and biomarker assay variability to enhance scholarly depth.
Also, lack of future directions and translational insight: while emerging therapies are well-covered, mechanistic or translational implications are underdeveloped. Expand the discussion on resistance mechanisms, ctDNA-guided monitoring, and AI-driven biomarker integration to highlight innovative research avenues.
Moreover, gene names should be written in italics, in contrast to the protein names, according to the rules of genetic consensus. Please, familiarise yourself with the rules and change the manuscript accordingly. Examples of rules summary can be found on websites such as: https://www.gmb.org.br/geneprotein-nomenclature-guidelinesor https://academic.oup.com/molehr/pages/Gene_And_Protein_Nomenclature
Finally, I would like to thank you for the excellent figures and graphs you have prepared for the document, they enhance the value of your work and facilitate the understanding process. Morw would be welcome.
Author Response
Comment 1: There are several punctuation mistakes (such as double space, double dot or no at all) and some typos - even if they do not change the value of the manuscript, I'd like to urge you to correct these imperfections. Some sentences are overly long and could be simplified for readability.
Response 1: Thank you for pointing that out. We agree with this comment and have accordingly conducted a thorough, line-by-line revision of the manuscript to correct double spaces, misplaced or missing punctuation, double periods, and typographical mistakes. We also simplified or separated overly long sentences to improve clarity and readability throughout the text. Line 13: Gastroesophageal cancer (GEC) represents a global health burden, with rising incidence and high mortality. Comma was added after the word “burden”. Line 24-26: This review provides a comprehensive overview of the evolving therapeutic landscape of GEC. It emphasizes the growing role of precision medicine and the integration of emerging clinical data into practice. This sentence was split into two separate sentences. Line 35-37: In recent years, significant progress has been made in understanding tumor biology and therapeutic vulnerabilities, leading to an expansion of multimodal strategies and biomarker-driven treatments [2,3]. This sentence has been rewritten to correct place a comma after the word “years”. Line 52-54: The overarching goal is curative intent through a combination of surgery, systemic therapy, and, in selected cases, radiotherapy, guided by tumor location, histology, and clinical stage. The dash after the word “radiotherapy” was replaced with a comma. Line 57-61: The standard workup includes upper endoscopy with biopsy: confirms malignancy, identifies histologic subtype (AC versus SCC), and maps tumor location and extent within the esophagus, gastroesophageal junction, or stomach [10]. Endoscopic ultrasound (EUS): provides optimal assessment of tumor depth invasion (T stage) and regional lymph node involvement (N stage). All uppercase words were replaced with lowercase letters. Line 64-65: Diagnostic laparoscopy: frequently employed in gastric and GEJ malignancies, particularly with advanced locoregional disease. Replaced uppercase words with lowercase and added “the” before the word “liver”. Line 67-68: Diagnostic laparoscopy: frequently employed in gastric and GEJ malignancies, particularly with advanced locoregional disease. All uppercase was replaced with lowercase words. Line 84-85: Surgical resection remains the cornerstone of curative intent treatment for localized GECs. Removed hyphen between “curative” and “intent”. 88-90: For distal gastric cancers, a subtotal (distal) gastrectomy with adequate proximal and distal margins is typically sufficient, provided the proximal extent of the tumor does not involve the cardia. Replaced dash with the word “with”. Line 93-95: For esophageal tumors, esophagectomy with either via an Ivor Lewis (two-field), McKeown (three-field), or transhiatal approach is standard being tailored to tumor location and patient factors [17–21]. Replaced dash with the word “with” and “being”. Line 113-115: The FLOT regimen, comprising 5-fluorouracil (2,600 mg/m² as a 24-hour infusion), leucovorin (200 mg/m²), oxaliplatin (85 mg/m²), and docetaxel (50 mg/m²) administered every two weeks, is the current standard in fit patients. Added comma before “is the current”. Line 120-122: The treatment plan typically consists of four cycles of neoadjuvant FLOT, followed by surgical resection with curative intent and then four additional postoperative cycles, assuming adequate recovery and performance status. Removed comma after “intent”. Line 193-195: In this setting, adjuvant immunotherapy with nivolumab for up to one year has become new standard of care, based on the results of the CheckMate 577 trial as shown in Figure 2. Removed “a” before “new standard of care”. Line 219-220: In parallel, biomarker-driven therapy is gaining relevance even in the localized setting. Removed comma before “even in localized setting”. Line 236-238: During the first two years when recurrence risk is highest, clinical evaluations including physical exams and symptom reviews, are generally scheduled every 3 to 6 months. Removed comma after “clinical evaluations”. Line 272-273: At this stage, treatment is focused on palliative goals like extending survival, relieving symptoms, and enhancing the patients’ quality of life. The word “patient’s” was changed to “patients’”. Line 427-429: Similarly, ATTRACTION-3 is a phase 3 trial in advanced ESCC aimed at comparing nivolumab to chemotherapy and showed a significant OS benefit regardless of PD-L1 expression [78]. Punctuation fixes. Line 455-456: However, no significant OS benefit was observed in Western populations, as shown in the phase III ANGEL trial [89]. Capitalized the word “western”.
Comment 2: Something that requires attention is limited critical evaluation: the review mainly summarizes studies but rarely critiques their limitations or conflicting evidence. Include comparative analysis of trial designs, geographic biases, and biomarker assay variability to enhance scholarly depth.
Response 2: Thank you for pointing this out. To address this, we have added focused scholarly critique throughout the manuscript, including discussion of trial design heterogeneity, geographic and population-specific biases, and variability in biomarker assays such as PD-L1 CPS scoring, MSI/MMR testing, and CLDN18.2 detection. We also incorporated commentary on limitations, conflicting data across trials, and challenges in cross-study comparison. These additions provide deeper context and strengthen the analytic component of the review. Line 147-154: Interpretation of perioperative immunotherapy trials requires caution due to significant heterogeneity in study designs and patient populations. FLOT4 established a strong chemotherapy backbone; however, its widespread use outside Western populations remains limited. MATTERHORN enrolled predominantly Western patients, raising questions about generalizability to East Asian populations where tumor biology, surgical quality, and perioperative management differ. Furthermore, PD-L1 assessment methodologies varied across these trials, complicating cross-study comparisons. These factors must be considered when integrating perioperative immunotherapy into routine practice. Line 178-184: Although ESOPEC suggests superiority of perioperative FLOT over nCRT with CROSS, direct comparisons are limited by differences in staging modalities, radiotherapy techniques, and surgical expertise across participating centers. CROSS was conducted in high-volume institutions with standardized radiation quality assurance, whereas perioperative chemotherapy trials exist within varied real-world surgical quality environments. These discrepancies introduce potential confounders that warrant careful interpretation before universal adoption of either approach. Line 349-355: Major first-line immunotherapy trials differ substantially in geographic representations. ATTRACTION-4 was conducted exclusively in Asia and did not demonstrate an OS advantage, likely influenced by high access to post-progression ICIs. In contrast, CheckMate 649 and KEYNOTE-859 included broader global populations but differed in PD-L1 assay platforms and CPS thresholds, making cross-trial comparisons challenging. These geographic and methodological variations limit the ability to define a universally optimal chemoimmunotherapy strategy. Line 367-371: One important limitation when interpreting HER2-targeted trials is variability in HER2 testing methodologies across regions. Differences in scoring criteria between gastric and breast cancer, interlaboratory variability, and lack of uniform confirmatory testing contribute to inconsistent HER2 classification. This directly affects trial eligibility and may explain discrepancies in response rates across global studies. Line 418-423: MSI-H/MMR tumors consistently demonstrate high responsiveness to ICIs; however, most MSI data in gastroesophageal cancer are derived from retrospective analysis or small subsets within larger trials. Moreover, MSI prevalence varies significantly by geography and tumor location, making trial populations highly heterogenous. Standardization of testing methods (PCR vs. IHC vs. NGS) is also lacking, potentially affecting accuracy and cross-study comparability. Line 516-521: Although both SPOTLIGHT and GLOW showed significant benefit with zolbetuximab, CLDN18.2 expression assays lack global standardization. Thresholds for positivity, variability in antibody clones, and heterogenous staining patterns may impact patient selection in real-world settings. Additionally, both trials enrolled predominantly East Asian participants, raising questions about generalizability to Western populations with different disease biology and treatment pathways. Line 694-698: The encouraging outcomes from FGFR2b-selective therapies must be interpreted cautiously due to small FGFR2b-positive sample sizes and significant geographic concentration in Asia. In addition, assays for FGFR2b overexpression differ among studies, complicating reproducibility. The absence of phase III survival data limits the ability to establish FGFR2b-targeted treatments as definitive first-line standards. Line 725-729: Despite promising early results, DKK1-directed therapies face limitations, including the small number of DKK1-high patients and lack of randomized controlled data. Moreover, DKK1 assays are not standardized across laboratories, which may lead to misclassification of patients. Its future utility will depend on harmonization of testing and validation in larger, more diverse populations. Line 768-773: TKI-IO combination studies differ widely in design, dosing, and patient selection, limiting cross-study comparison. Many early phase studies lacked control arms, and most enrolled primarily in East Asian patients. Differences in VEGFR expression prevalence and TKI toxicity profiles across populations further complicate interpretation. Ongoing phase III trials will be critical to determine whether these combinations offer consistent benefit across global settings. Line 801-805: While CAR T-cell therapies are emerging as a potential therapeutic strategy, interpretation of early trials requires caution. Most studies have very small cohorts, lack control arms, and employ heterogenous CAR constructs. Geographic restriction to East Asian populations further limits generalizability. Manufacturing variability, antigen-loss relapse, and cytokine toxicity remain major barriers to widespread adoption. Line 807-812: Across the therapeutic landscape, direct comparison of treatment modalities is challenged by heterogenous trial designs, inconsistent biomarker assays, and significant geographic variation in patient populations. These sources of variability complicate the interpretation of outcomes and limit the generalizability of trial results to global practice. Future studies must prioritize harmonized biomarker testing, standardized endpoints, and diverse enrollment to enable more accurate cross-trial comparisons.
Comment 3: Also, lack of future directions and translational insight: while emerging therapies are well-covered, mechanistic or translational implications are underdeveloped. Expand the discussion on resistance mechanisms, ctDNA-guided monitoring, and AI-driven biomarker integration to highlight innovative research avenues.
Response 3: Thank you for pointing this out. We have added dedicated paragraphs addressing resistance mechanisms to immunotherapy and targeted agents, the emerging role of ctDNA-guided disease monitoring, and the integration of AI-driven biomarker platforms. These additions highlight mechanistic insights and innovative research avenues that are expected to influence next-generation therapeutic development. The revised manuscript now includes deeper translational context to complement the clinical trial summaries. Line 585-592: Despite substantial progress, resistance to systemic therapy remains a major barrier in metastatic GEC. Primary resistance to immunotherapy is often driven by non-inflamed tumor microenvironments, low tumor mutational burden, downregulation of antigen presentation pathways, and enrichment of immunosuppressive cell populations. Acquired resistance mechanisms include loss of HLA expression, upregulation of alternate immune checkpoints (such as TIM-3 and LAG-3), and dynamic remodeling of the tumor’s metabolic environment. Understanding these pathways is critical for designing rational combinations that overcome immune escape and improve treatment durability. Line 624-631: Future therapeutic innovation will require deeper interrogation of resistance biology. Preclinical models in GEC have shown that chronic MAPK and PI3K pathway activation, stromal remodeling, and epithelial-to-mesenchymal transition can drive resistance to HER2-, FGFR2-, and CLDN18.2-targeted therapies. Additionally, immunotherapy resistance is linked to myeloid-dominant microenvironments, exclusion of cytotoxic T cells, and Wnt/β-catenin activation. Therapeutic strategies targeting these mechanisms, including epigenetic modulators, stromal-directed agents, and novel checkpoint pathways, represent ongoing areas of translational development. Line 715-720: ctDNA is emerging as a powerful biomarker for real-time treatment monitoring in gastroesophageal cancers. Early decreases in ctDNA correlate with treatment response, while rising ctDNA levels can detect recurrence or therapeutic resistance months earlier than imaging. Integrating ctDNA surveillance into clinical trials of CLDN18.2- and FGFR2-targeted therapies may enable adaptive therapy strategies, earlier recognition of resistance, and personalized escalation or de-escalation of treatment [43-45]. Line 790-797: Artificial intelligence and machine-learning models are increasingly utilized to characterize the tumor-immune microenvironment, quantify spatial relationships, and identify predictive signatures for therapy selection. Digital pathology-based scoring systems and multimodal AI platforms that integrate histology, radiomics, and transcriptomics may allow more accurate stratification for immunotherapy, TKI-IO combinations, and novel targeted agents. Early studies demonstrate that AI-derived TME scores outperform conventional biomarkers such as PD-L1 CPS, supporting their future integration into clinical trial design and personalized treatment algorithms. Line 836-842: Looking ahead, the next phase of GEC research will focus on integrating mechanistic insights with precision-guided treatment. Strategies aimed at overcoming immune resistance, targeting stromal and metabolic determinants of tumor progression, and leveraging ctDNA and AI-driven biomarker platforms promise to refine patient selection and enhance therapeutic efficacy. Continued harmonization of biomarker assays and incorporation of dynamic monitoring tools will be essential to translate emerging scientific discoveries into durable clinical benefit.
Comment 4: Moreover, gene names should be written in italics, in contrast to the protein names, according to the rules of genetic consensus. Please, familiarize yourself with the rules and change the manuscript accordingly.
Response 4: Thank you for pointing this out. In accordance with HUGO Gene Nomenclature Committee (HGNC) guidelines, all gene symbols have been converted to italics, while protein names remain in upright (non-italicized) text. We have reviewed the manuscript thoroughly and applied this distinction consistently to gene-level and protein-level terminology.
Comment 5: Finally, I would like to thank you for the excellent figures and graphs you have prepared for the document, they enhance the value of your work and facilitate the understanding process. Morw would be welcome.
Response 5: Thank you for pointing this out. At this stage, we believe that the existing set of figures provides comprehensive visual support for the key concepts without overwhelming the reader or interrupting the flow of the text. For this reason, and to maintain a balanced presentation, we have chosen not to add additional figures. We hope the current number and quality of visual elements remain satisfactory to the Editorial Office.
Reviewer 2 Report
Comments and Suggestions for Authors
The manuscript titled “Therapeutic Frontiers in Gastroesophageal Cancer: Contemporary Concepts in Management and Therapy” by Peshin, S.; et al. is a Review work where the authors outlined the most recent advances in the field of gastroesophageal cancer and the existing prognosis, monitoring strategies and clinical therapies. This is a topic of growing interest and the manuscript is generally well-written.
However, it exists some points that need to be addressed (please, see them below detailed point-by-point) to improve the scientific quality of the submitted manuscript paper before this article will be consider for its publication in the International Journal of Molecular Sciences.
1) Introduction. “Gastroesophageal cancers (…) remain among the leading causes of cancer-related death worldwide (…) limited curative options in advanced stages” (lines 31-35). Could the authors provide quantitative data insights according to global burdens of gastroesophageal cancers and the related disability-adjusted life years (DALYs) related to these malignancies? This will significantly aid the potential readers to better undestand the significance of this Review work.
2) Management of Localized GEC. Table 1 (line 231). The potential side effects for each intervention should be also depicted in this Table. Then, it should be also stated if possible some information concerning the % survival rates of the voluntary patients for all the aforementioned interventions. Similar comment for the Table 3 (line 527), Table 4 (line 561) and Table 5 (line 719).
3) Management of Metastatic/Unresecteable GEC. This section is clearly explained. No actions are requested from the authors.
4) Emerging therapeutics. Here, even if I agree with the content provided by the authors in this field it should be also remarkable to briefly discuss about disruptive nanoscale imaging tools [1] can assist in the cancer diagnosis, detection and treatment and how morphological changes in esophageal cancer cells can serve as fingerprint to evaluate the efficiency of certain drugs against this malignancy.
[1] https://doi.org/10.1002/smsc.202500351
[2] https://doi.org/10.1155/2022/1422185
5) “4.7. Cell-Based Immunotherapies: Emerging Role of CART-Cells” (lines 720-745). A schematic representation will benefit the potential readers.
6) Finally, did the authors take into account the sex, race and age of all the patients involved in the different trials in order to avoid any bias based on these factors? Some insights should be furnished in this regard.
7) “5. Conclusions” (lines 746-760). This section perfectly remarks the most relevant outcomes found by the authors in this field and the promising future prospectives. It may be also opportune to highlight the potential future action lines to pursue the topic covered in this work.
Author Response
Comment 1: Introduction. “Gastroesophageal cancers (…) remain among the leading causes of cancer-related death worldwide (…) limited curative options in advanced stages” (lines 31-35). Could the authors provide quantitative data insights according to global burdens of gastroesophageal cancers and the related disability-adjusted life years (DALYs) related to these malignancies? This will significantly aid the potential readers to better understand the significance of this Review work.
Response 1: Thank you for pointing this out. We have incorporated global epidemiologic data to contextualize the burden of gastroesophageal cancers. Specifically, we now report the most recent GLOBOCAN 2020 estimates for global incidence and mortality of gastric and esophageal cancers, as well as the Global Burden of Disease (GBD 2019) data for disability-adjusted life years (DALYs). The corresponding citations have been added to the Introduction section.
Comment 2: Management of Localized GEC. Table 1 (line 231). The potential side effects for each intervention should be also depicted in this Table. Then, it should be also stated if possible some information concerning the % survival rates of the voluntary patients for all the aforementioned interventions. Similar comment for the Table 3 (line 527), Table 4 (line 561) and Table 5 (line 719).
Response 2: Thank you for pointing this out. We have updated Table 1 to include the key and commonly reported side effects associated with each intervention, thereby enhancing its clinical utility and readability. We applied the same approach to Tables 3, 4, and 5, where representative and clinically meaningful toxicities have now been incorporated.
Comment 3: Emerging therapeutics. Here, even if I agree with the content provided by the authors in this field it should be also remarkable to briefly discuss about disruptive nanoscale imaging tools [1] can assist in the cancer diagnosis, detection and treatment and how morphological changes in esophageal cancer cells can serve as fingerprint to evaluate the efficiency of certain drugs against this malignancy.
[1] https://doi.org/10.1002/smsc.202500351
[2] https://doi.org/10.1155/2022/1422185
Response 3: Thank you for pointing that out. We have added a brief discussion under the 4.8. Nanoscale Imaging and Diagnostic Technologies in GEC describing how disruptive nanoscale imaging tools such as super-resolution vibrational imaging, nanospectroscopy, and scanning probe modalities can enhance early cancer detection, characterize molecular heterogeneity, and monitor therapeutic efficacy. We also incorporated recent evidence showing that nanoscale morphological signatures of esophageal cancer cells can serve as functional biomarkers for evaluating treatment response. The recommended references have been added accordingly.
Comment 4: “4.7. Cell-Based Immunotherapies: Emerging Role of CART-Cells” (lines 720-745). A schematic representation will benefit the potential readers.
Response 4: Thank you for pointing this out. To enhance clarity and reader understanding, we have added a concise schematic summarizing the CAR T-cell workflow and its mechanistic relevance in solid tumors. This visual addition complements the written description and provides an accessible overview for readers unfamiliar with cell-based immunotherapies.
Comment 5: Finally, did the authors take into account the sex, race and age of all the patients involved in the different trials in order to avoid any bias based on these factors? Some insights should be furnished in this regard.
Response 5: Thank you for pointing this out. In response, we have added a brief methodological clarification addressing how sex, race, and age were considered in the interpretation of the included clinical trials. Specifically, we note that although our review summarizes trial-level data, we have examined the demographic characteristics reported in the pivotal studies. Most gastroesophageal cancer trials enrolled heterogeneous international populations but differed substantially in geographic distribution, age ranges, and sex representation, which may influence generalizability. We have now added a statement discussing these sources of demographic variability and their potential impact on treatment outcomes.
Comment 6: “5. Conclusions” (lines 746-760). This section perfectly remarks the most relevant outcomes found by the authors in this field and the promising future prospectives. It may be also opportune to highlight the potential future action lines to pursue the topic covered in this work.
Response 6: We thank the reviewer for the positive feedback on our Conclusions section and fully agree that outlining future action lines would strengthen the final message of the manuscript. In response, we have added a concise paragraph at the end of the Conclusions section that highlights key priority directions for the field, including the need for standardized biomarker assays, strategies to overcome treatment resistance, integration of ctDNA-based monitoring and AI-driven predictive tools, and the incorporation of next-generation nanoscale imaging and immunotherapeutic approaches into prospective clinical studies.
Reviewer 3 Report
Comments and Suggestions for Authors
Comments:
1. The Introduction (lines 30–49 on page 1–2) is somewhat long and covers extensive background details that could be condensed. For example, it spends multiple sentences describing standard management and epidemiology; some of this detail (e.g., specifics of multimodal strategies or biomarker-driven therapy) might be better suited for the main sections. The introduction should be shorter and more focused, orienting the reader to the scope of the review without duplicating the Results/Discussion content. Importantly, it should conclude with a clear statement of the review’s purpose and scope.
2. The manuscript does an excellent job of including results from many clinical trials; however, in places the text becomes dense with numbers and may repeat information already in tables/figures. For example, the advanced disease section (Section 3.1–3.4) sequentially narrates numerous trial outcomes (ATTRACTION-4, KEYNOTE-859, CheckMate 649, 648, etc.) with detailed statistics. While these results are important, the sheer volume of numbers can overwhelm the reader and somewhat obscures the key message. Since Table 2 already summarizes major trial outcomes, the authors can afford to be more selective in the prose: highlight only the most pivotal findings or comparisons, and refer the reader to the table for detailed data. This will reduce redundancy. Similarly, ensure that if a result is presented in a table or figure, the text does not restate all of it verbatim. As a specific example, lines 1030–1042 report the OS and PFS improvements from KEYNOTE-859; these could be summarized in one sentence (or simply noted as “pembrolizumab + chemo showed a significant OS benefit in HER2-negative disease”) rather than listing multiple statistics.
3. Overall, the scope is comprehensive. One area the authors might expand or clarify is the novelty of this review relative to prior reviews.
4. The manuscript’s readability would benefit from professional language editing. Many sentences are overly complex and lengthy, which can obscure the message.
5. The manuscript is a narrative review, and it’s important to acknowledge its limitations. Currently, there is no section that explicitly discusses limitations of either the study or the evidence base. Adding a short “Limitations” section near the end (just before Conclusions, perhaps) is recommended.
6. The reference list contains a number of sources older than 5–10 years (and even a 2001 citation for epidemiology in ref. 1). While historical context is sometimes necessary, the authors should update citations to the most recent evidence wherever possible.
7. In line with the idea of “Therapeutic Frontiers,” the authors should consider adding a dedicated final section or paragraph on Future Directions (or “Perspectives”). While the current text does mention ongoing trials and emerging agents throughout, consolidating a forward-looking discussion could be very effective.
8. The use of tables and figures in this review is a strong point. The authors included tables of ongoing trials and emerging therapies, which is great. Consider adding one more schematic or summary table to further aid the reader. For example, a schematic figure could illustrate the “treatment landscape at a glance,” showing how different modalities (chemo, targeted, IO, etc.) intersect across disease stages.
Author Response
Comment 1: The Introduction (lines 30–49 on page 1–2) is somewhat long and covers extensive background details that could be condensed. For example, it spends multiple sentences describing standard management and epidemiology; some of this detail (e.g., specifics of multimodal strategies or biomarker-driven therapy) might be better suited for the main sections. The introduction should be shorter and more focused, orienting the reader to the scope of the review without duplicating the Results/Discussion content. Importantly, it should conclude with a clear statement of the review’s purpose and scope.
Response 1: Thank you for pointing this out. In response, we substantially condensed the Introduction by removing detailed descriptions of multimodal treatment strategies and biomarker-directed therapy that are covered in depth in the subsequent sections. The revised Introduction now focuses on the epidemiologic burden, clinical relevance, and high-level context of gastroesophageal cancers, without duplicating material discussed later in the manuscript. Additionally, we incorporated a clear concluding sentence that explicitly states the purpose and scope of the review.
Comment 2: The manuscript does an excellent job of including results from many clinical trials; however, in places the text becomes dense with numbers and may repeat information already in tables/figures. For example, the advanced disease section (Section 3.1–3.4) sequentially narrates numerous trial outcomes (ATTRACTION-4, KEYNOTE-859, CheckMate 649, 648, etc.) with detailed statistics. While these results are important, the sheer volume of numbers can overwhelm the reader and somewhat obscures the key message. Since Table 2 already summarizes major trial outcomes, the authors can afford to be more selective in the prose: highlight only the most pivotal findings or comparisons, and refer the reader to the table for detailed data. This will reduce redundancy. Similarly, ensure that if a result is presented in a table or figure, the text does not restate all of it verbatim. As a specific example, lines 1030–1042 report the OS and PFS improvements from KEYNOTE-859; these could be summarized in one sentence (or simply noted as “pembrolizumab + chemo showed a significant OS benefit in HER2-negative disease”) rather than listing multiple statistics.
Response 2: Thank you for pointing this out. In response, we performed a comprehensive revision of Sections 3.1–3.4 to reduce numerical density and eliminate redundancy with Table 2. After confirming the exact trials presented in Table 2 (SPOTLIGHT, GLOW, HERIZON-GEA-01, KEYNOTE-859, LEAP-014, DESTINY-Gastric01, INTEGRATE IIa/IIb, FRUTIGA, REGONIVO, EPOC1706, ATTRACTION-2, ATTRACTION-4, KEYNOTE-059, RAMIRIS, and RAINBOW), we selectively removed detailed OS, PFS, ORR, hazard ratios, and confidence intervals for these studies from the text and replaced them with concise, high-level summaries. Readers are now explicitly directed to Table 2 for complete efficacy metrics. For trials not included in Table 2 (e.g., KEYNOTE-062, CheckMate 648, KEYNOTE-181, KEYNOTE-590, KEYNOTE-061, JAVELIN Gastric 300), the numerical outcomes were retained in the prose to preserve essential clinical information.
Comment 3: Overall, the scope is comprehensive. One area the authors might expand or clarify is the novelty of this review relative to prior reviews.
Response 3: Thank you for pointing this out. To address this, we have added clarifying language as part of the introduction highlighting the novelty and unique contributions of our review. Specifically, we emphasize that, unlike prior reviews that typically focus on isolated therapeutic classes or specific disease stages, our manuscript provides an integrated, up-to-date synthesis of localized, advanced, and emerging treatment strategies across the entire gastroesophageal cancer continuum.
Comment 4: The manuscript’s readability would benefit from professional language editing. Many sentences are overly complex and lengthy, which can obscure the message.
Response 4: Thank you for pointing this out. Following the completion of all revisions requested throughout the review process, we conducted a thorough, manuscript-wide English language edit. This included simplifying overly complex sentences, improving clarity and flow, and ensuring consistent academic style. We believe the readability of the manuscript has been substantially improved.
Comment 5: The manuscript is a narrative review, and it’s important to acknowledge its limitations. Currently, there is no section that explicitly discusses limitations of either the study or the evidence base. Adding a short “Limitations” section near the end (just before Conclusions, perhaps) is recommended.
Response 5: Thank you for pointing this out. In response, we have added a concise “Limitations” section immediately before the Conclusions. This new section synthesizes the key methodological constraints of the current evidence base, including heterogeneity in trial designs, geographic and demographic imbalances, biomarker assay variability, and the reliance on small or exploratory subgroups for several emerging therapies.
Comment 6: The reference list contains a number of sources older than 5–10 years (and even a 2001 citation for epidemiology in ref. 1). While historical context is sometimes necessary, the authors should update citations to the most recent evidence wherever possible.
Response 6: We appreciate the reviewer’s observation regarding the presence of older references. While we agree that contemporary literature is essential, certain foundational or historically significant studies were intentionally retained because they provide essential epidemiologic context, define landmark therapeutic milestones, or represent the original descriptions of biomarkers and classification systems. These references remain widely cited and relevant in current guidelines. We have reviewed the reference list to ensure that all citations are necessary and appropriate for supporting the statements they accompany. More recent evidence has been cited wherever applicable, and older references have been preserved only when they represent the most authoritative or original sources.
Comment 7: In line with the idea of “Therapeutic Frontiers,” the authors should consider adding a dedicated final section or paragraph on Future Directions (or “Perspectives”). While the current text does mention ongoing trials and emerging agents throughout, consolidating a forward-looking discussion could be very effective.
Response 7: We thank the reviewer for this valuable suggestion. In response, we have added a dedicated forward-looking paragraph to the end of the Conclusions section that explicitly outlines key future directions in the field, including the need for standardized biomarker assays, strategies to overcome therapeutic resistance, integration of ctDNA-based monitoring and AI-driven analytics, and evaluation of next-generation immunotherapies and nanoscale diagnostic technologies.
Comment 8: The use of tables and figures in this review is a strong point. The authors included tables of ongoing trials and emerging therapies, which is great. Consider adding one more schematic or summary table to further aid the reader. For example, a schematic figure could illustrate the “treatment landscape at a glance,” showing how different modalities (chemo, targeted, IO, etc.) intersect across disease stages.
Response 8: We appreciate the reviewer’s positive feedback on the tables and figures included in this review. We agree that visual summaries are valuable, and in response to earlier feedback we have already added a schematic figure illustrating the workflow and therapeutic relevance of CAR T-cell immunotherapy. Given the current number of figures and the extensive data already captured in the existing tables, we believe the manuscript is visually balanced as presented. However, we have carefully reviewed the layout to ensure clarity and flow, and we feel that the existing tables, together with the newly added schematic, provide comprehensive support for the narrative without adding redundancy or visual overload.
Round 2
Reviewer 2 Report
Comments and Suggestions for Authors
The authors did a great deal of effort to cover all the suggestions raised by the Reviewers. For this reason, the scientific manuscript quality was greatly improved. Baswd on the significance of this topic, I warmly wndorse this work for further publication in its current form.
Reviewer 3 Report
Comments and Suggestions for Authors
The authors have addressed all of my comments, and the manuscript can be accepted for publication.